# Abundance does not predict extinction risk in the fossil record of marine plankton

Sarah Trubovitz [1,3✉], Johan Renaudie [2✉], David Lazarus[2] & Paula J. Noble [1]

A major premise of ecological neutral theory is that population size is inversely related to extinction risk. This idea is central to modern biodiversity conservation efforts, which often rely on abundance metrics to partially determine species extinction risk. However, limited empirical studies have tested whether extinction is indeed more probable for species with low abundances. Here we use the fossil record of Neogene radiolaria to test the relationship between relative abundance and longevity (time from first to last occurrence). Our dataset includes abundance histories for 189 polycystine radiolarian species from the Southern Ocean, and 101 species from the tropical Pacific. Using linear regression analyses, we show that neither maximum nor average relative abundance are significant predictors of longevity in either oceanographic region. This suggests that neutral theory fails to explain the plankton ecological-evolutionary dynamics we observe. Extrinsic factors are likely more important than neutral dynamics in controlling radiolarian extinction.

[1] Department of Geological Sciences & Engineering, University of Nevada - Reno, Reno, NV, USA. [2] Museum für Naturkunde, Leibniz-Institut für Evolutions- und Biodiversitätsforschung, Berlin, Germany. [3] Present address: Department of Biological Sciences, University of Southern California, Los Angeles, CA, USA. ✉email: trubovit@usc.edu; johan.renaudie@mfn.berlin

Understanding the patterns and correlates of extinction is of great importance to modern conservation biology as well as our interpretations of the fossil record throughout the history of life. There has been much debate as to whether extinction impacts taxa differently based on their unique traits and life strategy, or if instead extinction is a stochastic process that affects all taxa equally, regardless of their biology and ecology[1–4]. Previous studies have found links between extinction risk and geographic range size[5–7], population trends[8], body size[9], and various reproductive and life history traits[10]. Higher taxonomic and morphology-based differences in longevity have also been observed[11,12]. But perhaps one of the most widely discussed correlates of extinction is taxon abundance[2,5,10,13–15]. The idea that rare taxa are more likely to go extinct than common taxa is an important tenant of ecological theory[16,17], and is among the core factors used to classify extinction risk in the conservation of modern species[18]. While extinction patterns in the past may not be perfect analogs for those presently underway becuase of the novel threats of anthropogenic activity[10], the fossil record of past extinction events nonetheless provides vital insight into macroecological and evolutionary processes associated with extinctions, and may help us identify the taxa most at risk of extinction today[16,19]. One way of exploring extinction risk in the fossil record is to examine species longevity from origination to extinction; species with lower abundances should have been at greater risk of extinction, and thus on average have shorter temporal ranges. Here we use the stratigraphic ranges of Neogene polycystine radiolarian species from the Southern Ocean (SO) and eastern equatorial Pacific (EEP) to test for a relationship between species abundance and longevity. We also test whether species are at differential risk of extinction based on their order-level taxonomic classification or biogeographic range category (endemic versus cosmopolitan).

The existing paradigms for explaining species abundance, diversity, and extinction patterns can be separated into two basic categories: niche theory and neutral theory. Niche theory is centered on the idea that each species fulfills a unique role in its environment, exhibiting a specific set of traits that influence its probabilities of survival and extinction[20]. By contrast, neutral theory aims to explain patterns in nature by assuming that all individuals of a given trophic level have identical odds of survival to reproduction and death, no matter which species they belong to[1,17]. Most published tests of neutral theory have focused primarily on modern species abundance distributions[17,21], but neutral theory also has important implications for extinction that have been largely unexplored in paleontological research[16]. It follows from neutral theory that there is an inverse relationship between a taxon's population size and probability of extinction, because purely stochastic processes make it more likely for rare species to random walk into extinction (population size = 0) than for common species. Therefore, if extinction is truly neutral and nonselective of species traits, we would expect to find a positive correlation between taxon abundance and longevity (the amount of time it persisted from origination to extinction). Conversely, if extinction is a selective process, we would not expect abundance to necessarily correlate with species longevity, and instead longevity would be related to specific biological traits and ecological roles of individual taxa.

The present study aims to contribute to our understanding of extinction risk factors using the fossil record of Radiolaria. Radiolarians are an ecologically important and diverse group of planktonic protists[22–24] that inhabit much of the ocean water column from the tropics to poles, and occupy a variety of niches as herbivores, predators, detritovores, and hosts of photosynthetic symbionts that contribute substantially to primary production[25]. Polycystine radiolarians build complex siliceous skeletons which readily fossilize and become a major component of the marine sediment record[26], making them a useful group for studying evolutionary trends in marine plankton[24,27,28]. Recent work has found that radiolarian species richness has remained relatively high and stable in the low latitudes over the last 10 million years, whereas high latitude radiolarians experienced a spike in extinction rates and decline in richness over the last 5 million years[27] (Fig. 1). In this study, we utilize comprehensive radiolarian species occurrence and relative abundance datasets from the Southern Ocean (22 Ma – Recent) and eastern equatorial Pacific (10 Ma – Recent), to determine whether relative abundance is related to stratigraphic longevity. Simple linear regression models are used to assess whether maximum or average (mean) relative abundance are important predictors of longevity, directly testing the assumption in ecological neutral theory that these two variables are related. To explore the alternative possibility that extinction is primarily driven by nonrandom biological factors, we also test whether higher-level taxonomic identity or biogeographic range size are predictors of species longevity using linear mixed effects models and analysis of variance (ANOVA) tests.

Our study utilizes extensive, quantitative species-level abundance count data (~2,500+ individuals per sample), is at high temporal resolution (~0.5–1 my sampling intervals for the tropical Pacific and ~0.1–1 my intervals for the Southern Ocean), and includes reasonably-complete sampling of species geographic ranges given their broad spatial distribution patterns[27,29,30]. Thus, the study system and attributes of our dataset make it well-suited for a robust test of neutral dynamics in a large number of species within an ecologically important clade. Our results challenge the ability of neutral theory to explain plankton dynamics, and raise questions as to the relevance of rarity as an indicator of extinction risk, which may inform modern species conservation practices.

## Results

**Relative abundance as a predictor of longevity**. The species in our study spanned four orders of magnitude in mean relative abundance (in our data, a good proxy for absolute abundance - see Methods) and three orders of magnitude in maximum relative abundance. Mean relative abundance of species ranged from 0.001% to 13%, and maximum relative abundance ranged from 0.01% to 45% (Supplementary Data 1). In the Southern Ocean (SO), the median average relative abundance was 0.038%, and the median value of maximum relative abundance was 0.315%. Species longevity varied between 0.5 and 20.9 million years in the SO, with a median value of 8.2 million years. In the eastern equatorial Pacific (EEP), the median average relative abundance was 0.044%, and the median value of maximum relative abundance was 0.093%. Species longevity ranged from 1.0 to 7.5 million years in the EEP, with a median value of 3.5 million years. None of our regression analyses revealed a clear relationship between longevity and either metric of relative abundance (mean or maximum) for either dataset (Fig. 2). Each regression yielded an R-squared value < 0.1, indicating that the linear models fit the data very poorly. For average relative abundance as a predictor of longevity, the lack of a relationship between these variables was confirmed by insignificant $p$ values ($p \gg 0.05$) in both the SO and EEP datasets. Despite showing very poor fits to the data, the linear models for maximum relative abundance as a predictor of longevity did produce significant $p$ values ($p < 0.05$) for both datasets. Given the low R-squared values of these regressions, however, we do not interpret this to be a meaningful result because it has virtually no predictive power. Visual assessment of the data confirms that longevity is not clearly correlated, linearly or otherwise, with either metric of abundance. Residuals plots for all four regression analyses also suggest that there is no nonlinear

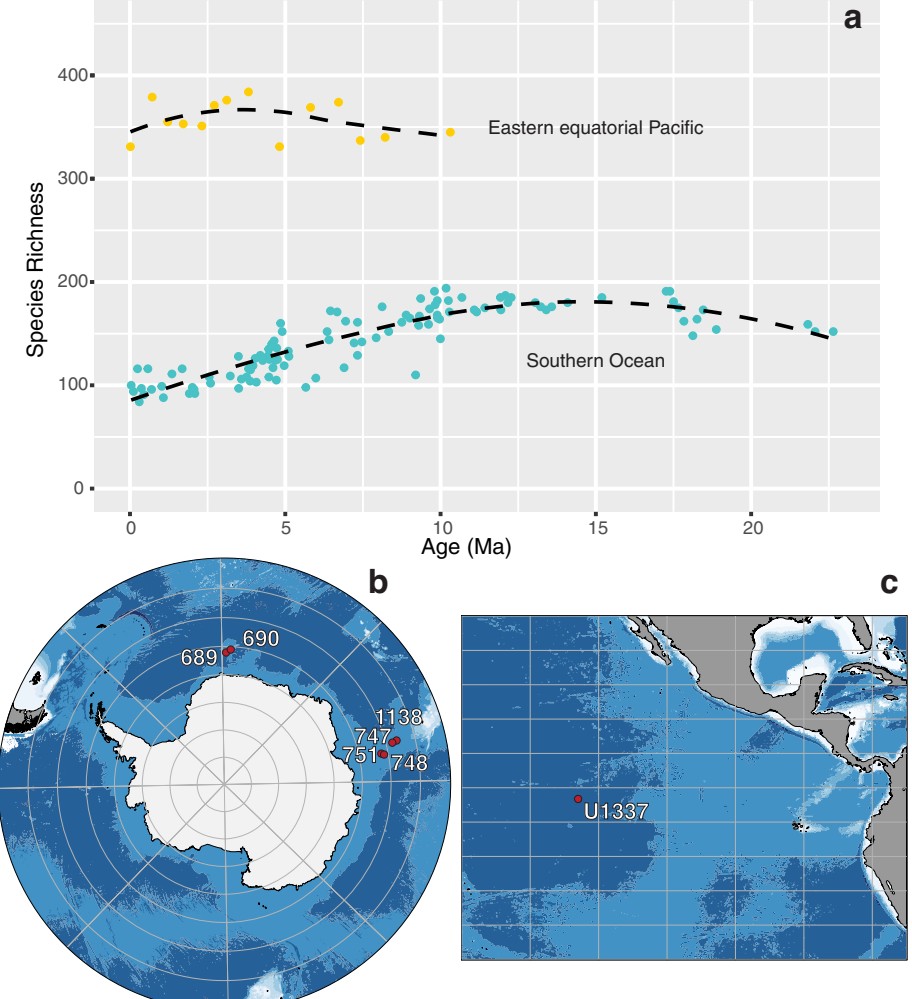

**Fig. 1 Species richness trends over time and sample localities in the eastern equatorial Pacific (EEP) and the Southern Ocean (SO).** Each point in (**a**) represents the raw species richness documented from a dated sediment core sample. Species richness data are from refs. [27,28]. SO data are combined from IODP (and predecessor programs) sites 689, 690, 747, 748, 751, and 1138 (**b**). EEP data come from IODP site U1337 (**c**). Maps depict the present-day locations of ocean drill-core sites used in this study. Basemaps were generated with R package ggOceanMaps[74].

relationship between the independent and dependent variables, and that the datasets generally meet the assumptions of regression models (Supplementary Fig. 1). Standard deviation in relative abundance was also tested as a predictor for longevity using linear regression models, but like the other regression models, no clear relationship was observed (Supplementary Fig. 2). The weak positive trend shown in the EEP standard deviation regression (R-squared = 0.12) is driven by five outlier species, which each occur only in two sequential samples and have the same discrete number of counted specimens in each sample (see Supplementary Data 1). Therefore, the short stratigraphic range of these species is partly related to their atypically low standard deviation in relative abundance. The weak positive trend (R-squared = 0.1) is thus not interpreted to be representative of the data. Overall, our results show that neither mean, maximum, nor variation in relative abundance were strong predictors of extinction risk in either oceanographic region.

**Other predictors of longevity.** To explore a potential link between higher taxonomic identity and longevity for radiolarians, we examined the frequency distribution of longevities for the three taxonomic orders of polycystine radiolarians included in our SO and EEP datasets (Fig. 3). Each dataset was treated

separately since the time interval covered and frequency between samples was different (see Methods). In the SO, the distribution of species longevities we observed was similar to the only previous study on this fauna[31], which included a much smaller number of taxa from the same time period and region. Nassellarians ($n = 135$) had an average longevity of 8.39 my, spumellarians ($n = 39$) averaged 9.62 my, and collodarians ($n = 15$) had an average of 6.62 my (although, with only 15 collodarian taxa in the dataset this is not a very robust estimate of their average longevity). The distribution of longevity data indicates no clear difference between the longevities of species in any of the three polycystine radiolarian orders (Fig. 3a). An ANOVA test confirmed that there is no statistically significant difference between them (F = 2.041, $p = 0.133$). Although it is possible that in the SO collodarians are slightly shorter-lived, and spumellarians are slightly longer-lived than nassellarians, it is difficult to generalize given that nassellarians had much better representation in our dataset than the other two groups.

In the EEP, mean species longevity was much shorter than in the SO (3.68 my in the EEP versus 8.51 my in the SO). The shorter length of time represented by the EEP dataset has biased it toward representing the short-ranging taxa; however, comparisons between taxonomic groups during this time interval are

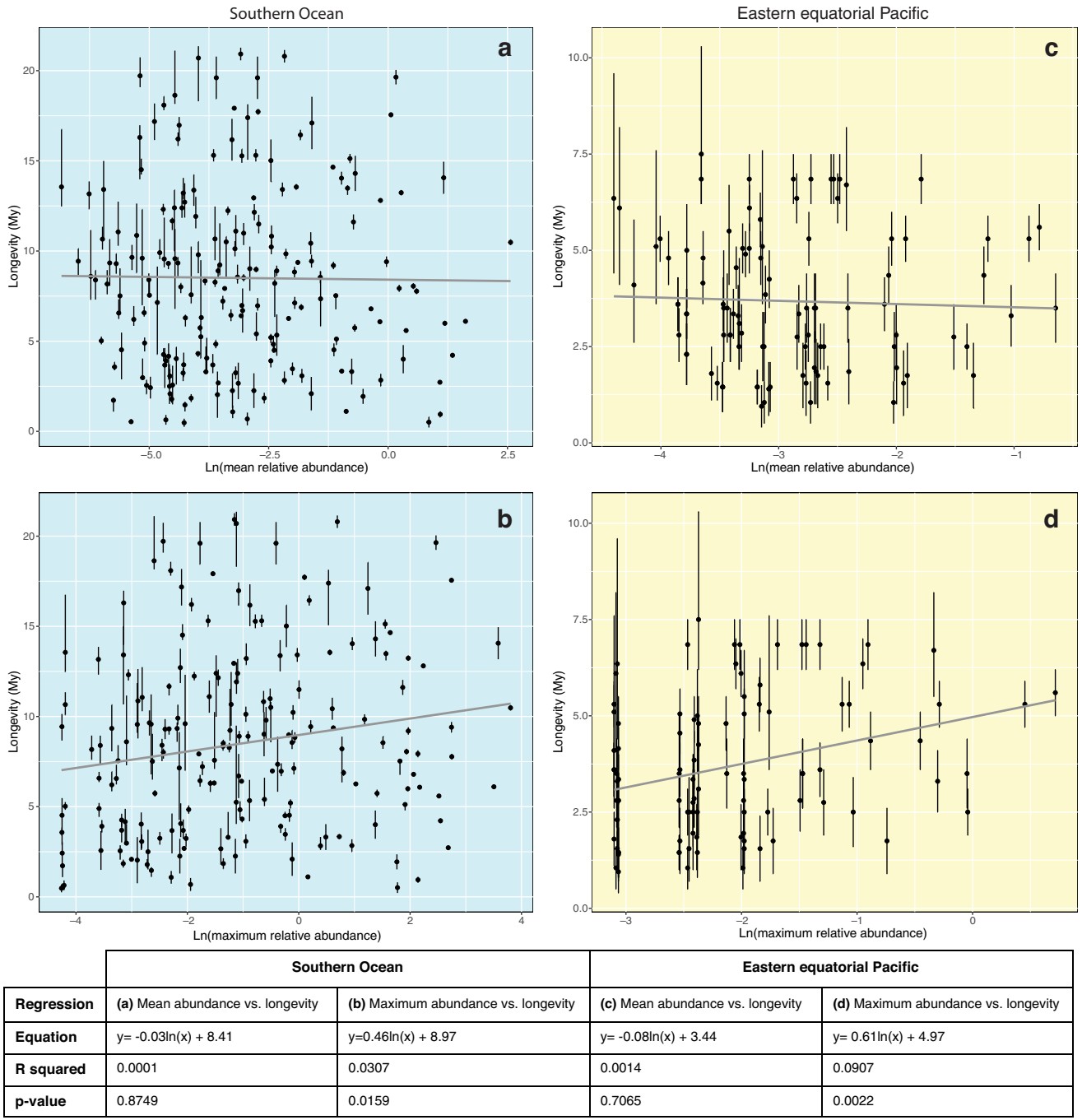

| Regression | **Southern Ocean** | | **Eastern equatorial Pacific** | |
|---|---|---|---|---|
| | **(a) Mean abundance vs. longevity** | **(b) Maximum abundance vs. longevity** | **(c) Mean abundance vs. longevity** | **(d) Maximum abundance vs. longevity** |
| **Equation** | y= -0.03ln(x) + 8.41 | y=0.46ln(x) + 8.97 | y= -0.08ln(x) + 3.44 | y= 0.61ln(x) + 4.97 |
| **R squared** | 0.0001 | 0.0307 | 0.0014 | 0.0907 |
| **p-value** | 0.8749 | 0.0159 | 0.7065 | 0.0022 |

**Fig. 2 Regression analyses comparing longevity to relative abundance for radiolarian assemblages from the Southern Ocean (SO) and eastern equatorial Pacific (EEP).** Mean relative abundance (independent variable) versus longevity (dependent variable) is shown for the SO in (**a**) and for the EEP in (**c**). Maximum relative abundance (independent variable) versus longevity (dependent variable) is shown for the SO in (**b**) and for the EEP in (**d**). For the SO regressions, $n = 189$ species. EEP regressions are based on $n = 101$ species. No metric of abundance showed a strong relationship to longevity in either dataset (R-squared <0.1 in all four regressions). Regressions using mean abundance as the independent variable were not statistically significant ($p \gg$ 0.05). The very weak positive relationship between maximum abundance and longevity was statistically significant ($p < 0.05$) in both the SO and the EEP. However, given the low R-squared values for these regressions, the data are clearly not well-fit and the regression has effectively no predictive power. Abundance metrics are normalized using a natural logarithm (ln) of the raw values, as explained in Methods. Error bars on longevity are designated for each species based on average gap size between occurrences (see Methods and Supplementary Data 1).

valid even if they cannot be directly integrated with the SO dataset. Nassellarians ($n = 58$) had an average longevity of 3.80 my, spumellarians ($n = 31$) averaged 3.58 my, and collodarians ($n = 12$) had an average of 3.34 my (Fig. 3b). Like the SO results, collodarians tended to be the shortest-lived species, but not significantly so and the pattern is likely in part due to their

relatively small portion of the dataset (only 12 species were observed to have their full ranges within the EEP study interval). The EEP nassellarians were slightly longer lived than the spumellarians, in contrast to the SO findings, but it does not approach a significant difference (ANOVA; $F = 0.399$, $p = 0.672$). Thus, each of our regional datasets independently suggest that

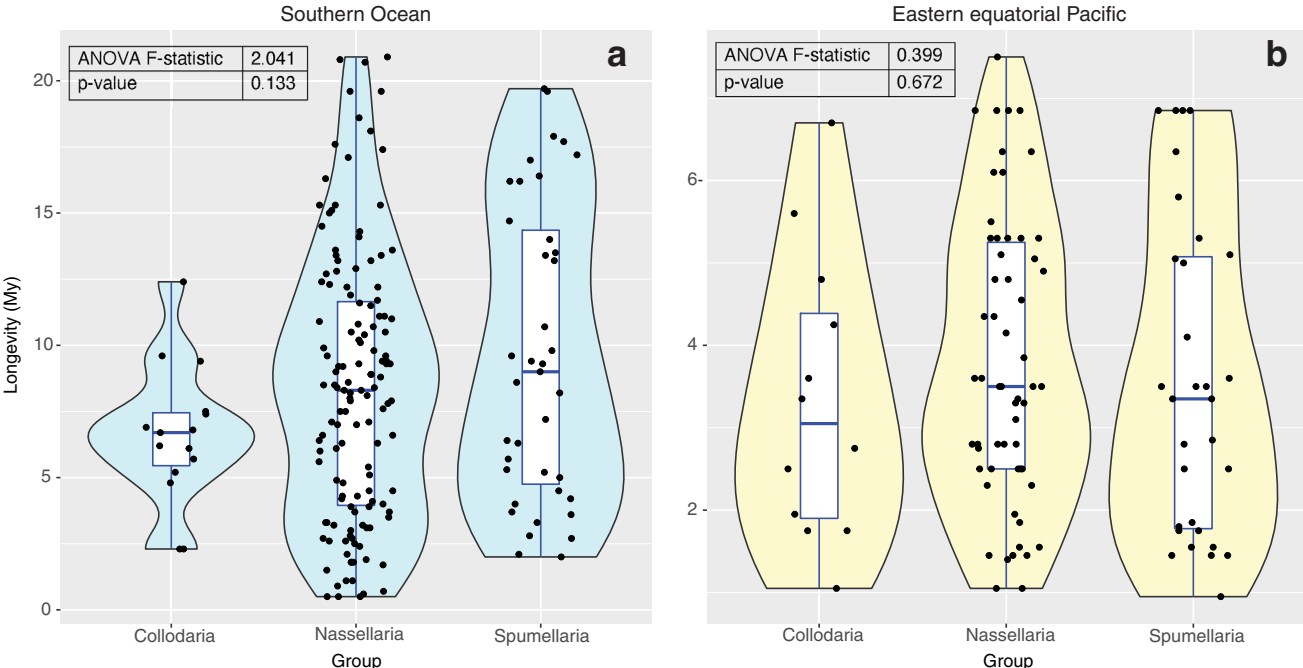

**Fig. 3 Distribution of species longevities among major taxonomic groups.** Violin plots illustrate the density of longevity values for Collodaria, Nassellaria, and Spumellaria in the SO (**a**) and EEP (**b**). Embedded box plots indicate the median, interquartile range, minimum and maximum longevity value for species within each taxonomic group. Each black point represents an individual species in the dataset. $N = 189$ species in the SO analysis and $n = 101$ species in the EEP analysis. All data underlying these plots are in Supplementary Data 1. ANOVA test results are given in the top left corner of each panel. With $p$ values » 0.05, the ANOVA tests confirm that there is no statistically significant difference between the longevities of the three polycystine radiolarian orders in either oceanographic region.

there is no difference in species longevities for the three orders of polycystine radiolarians.

To investigate whether geographic range size influences species longevity in our dataset, the stratigraphic ranges of cosmopolitan versus endemic species were compared. Species in each oceanographic region were classified as cosmopolitan if they occurred in both the EEP and SO, or endemic if they were only observed in one region and not the other. These broad categories are not meant to fully represent species biogeographic ranges, but they do allow us to explore the potential role of biogeography in predicting longevity despite the absence of more precise radiolarian species range data. Linear mixed effects models were designed to account for both mean relative abundance and biogeographic classification as a joint predictor for longevity. These models show no significant relationship between variables (Supplementary Fig. 3). Furthermore, the linear mixed effects models that include both average relative abundance and biogeographic range are marginally worse predictors of longevity than those utilizing abundance metrics as the sole predictor (for the EEP dataset, $\Delta AICc = 5.4$; for the SO dataset, $\Delta AICc = 4.1$). Box plots comparing the longevities of cosmopolitan and endemic species in each region also show no difference based on biogeographic range (Supplementary Fig. 3). ANOVA test results confirm that there is no significant difference between the longevity of cosmopolitan versus endemic species in either region (SO dataset: $F = 1.34$, $p$ value $= 0.25$; EEP dataset: $F = 2.18$, $p$ value $= 0.14$). We were therefore unable to identify any underlying general species characteristics contributing to extinction risk.

## Discussion
Ecological neutral theory predicts a negative relationship between abundance and extinction risk. Rare taxa should be more at risk of extinction, and thus have shorter longevities. However, our

data did not support this expectation. Instead, we found that maximum and average relative abundance had no effect on species longevity for Neogene – Recent radiolarian species from two regions of the pelagic ocean: the Southern Ocean (SO) and eastern equatorial Pacific (EEP). It appears that more abundant taxa were not proportionally buffered against extinction, and rare taxa were not predisposed to short durations (Fig. 2). None of the regression analyses found a significant correlation between any metric of relative abundance (mean, maximum, or standard deviation) and species longevity. Not only did regression analyses yield nonsignificant or extremely weak results, but no other hint of a relationship (e.g., nonlinear) between abundance and longevity can be discerned visually from the plots (Fig. 2) or from the distribution of regression residuals (Supplementary Fig. 1).

The fact that the SO and EEP datasets yielded the same negative result is particularly surprising when we consider their strikingly different macroevolutionary contexts. Over the last 10 million years (my), species richness and community structure (evenness) in the EEP were very stable, suggesting that only normal background extinction was taking place[27] (Fig. 1). By contrast, the SO experienced a profound decline in radiolarian species richness over the last 5 my, following a disruption in community structure, both of which have been interpreted as responses to intense regional climate cooling[27]. While this pattern is not part of a recognized mass extinction event, extinction rates were elevated well above normal background levels in the SO (12.5% of species in each time bin went extinct over the last 5 my, compared with a 3.2% extinction rate from 22-5 Ma)[27]. Despite these different modes of extinction, both datasets indicate that longevity is not tied to abundance.

Previous paleontological research on the relationship between abundance and extinction have primarily used the marine macroinvertebrate record[5,8,13–15,32,33] and have produced mixed results. Some studies found clear relationships between

abundance and longevity[8,13,14,32] while another showed that geographic range was a better predictor than abundance[5]. Meanwhile, during mass extinction events, it appears that high abundance does not promote survivorship[15,33]. There are relatively few studies that examine extinction selectivity in marine plankton. Among the planktonic foraminifera, Neogene globorotaliid and globogerinid lineages exhibit significantly different longevities, and lineages classified as "dominant" had longer durations than less abundant taxa[12]. The results of this study can be interpreted to support niche theory or neutral theory, since they show that both taxonomic identity (i.e., unique traits) and abundance (which can be related to neutral dynamics or to unique traits) were each significantly linked to longevity. Another study found that Cretaceous-Recent planktonic foraminifera species with globose tests were less likely to go extinct than species with keeled tests[11], indicating that extinction selects for certain morphologies and is not a random process, as neutral theory posits.

It is important to consider that plankton population sizes are so much larger than most other organisms that they may alter predictions from neutral theory. The stochastic demographic parameters of neutral theory (random survival, reproduction, and death of individuals) produce species longevities for the plankton that are unrealistic[34]. Under solely neutral dynamics, abundant species could theoretically persist for hundreds of millions to billions of years, while the micropaleontological record shows that protistan plankton species durations are on the order of ~5–15 million years on average[11,31,35]. Allen and Savage[34] addressed this problem by adjusting the neutral model to add a parameter for the combined effects of demographic and environmental stochasticity, which allows environmental change to impact population sizes as well as neutral demographic behavior. In addition, Allen and Savage's model creates variation in abundance on a per-capita basis, instead of on a purely individual level[34], which prevents abundant species from being virtually immune to individual-based random walk and consequent extremely long species durations. Despite these improvements, Allen and Savage's model shares a basic assumption with the original neutral theory: abundance and longevity must be positively related.

The lack of a strong relationship between abundance and longevity in our study agrees with the findings of some macroinvertebrate analyses[5,15,33]. Two of these studies, however, were explicitly investigating survivorship across mass extinction intervals[15,33], which are times when evolutionary processes are thought to be fundamentally different from typical background extinction dynamics[36]. Our results contrast with other macroinvertebrate studies[8,13,14,32], all of which showed a relationship between longevity or extinction risk and abundance. Our incongruent results could be partially due to the differences in the structure of the datasets used. This radiolarian dataset is comprised of species-level count data, rather than genus or subgenus-level taxon occurrences (as in refs. [13–15,33]). Speciation and extinction occur on the species level, so these processes can only be estimated, not measured, using genera[37]. Thus, it is not clear the extent to which the patterns observed in previous studies would scale to species-level data on the same macroinvertebrate study systems. The temporal resolution of our radiolarian data (~0.1–1 my) is much higher than in refs. [8,13,14], which may also have contributed to our conflicting results. Coarse temporal resolution mixes data from many paleocommunities, which could have had very different taxon relative abundances, and combine occurrences of taxa that never coexisted in nature. Therefore, the choice of temporal bin length can affect the results of an abundance-extinction correlation test, and very long bins used in these macroinvertebrate studies (on the order of geologic stages or ~10 my) likely dilute or distort the signal being studied. Lastly,

all previous studies have either used indirect proxies for abundance (e.g., number of locations reporting the presence of a species) rather than actual abundance data, or have used abundance data from small samples (mostly <300 individuals) which prevent relative abundance estimation for more than a handful of the most common species. Given these major differences in data structure, it is only possible to make broad comparisons between studies.

Why might our results deviate from the relationship predicted by neutral theory? The data used to develop and refine the unified neutral theory in ecology are almost entirely based on extant, large terrestrial organisms, such as forest and avian datasets[38–41]. Plankton populations differ from these study systems in that local assemblages are very well mixed on a global scale via ocean currents, so dispersal capability is probably less important than environmental preference in determining species local abundances[24]. This fundamentally differs from the limitations on the dispersal of large terrestrial organisms, like trees and birds, and some marine macroinvertebrates. Our results suggest that radiolarian species are indeed capable of persisting for long periods of time at relatively low abundances, which is a phenomenon that could be related to their dispersal and life history traits, although our data do not permit us to test this idea. The prevalence of relatively rare species in modern microbial communities has been observed as well[42], suggesting that rarity could be a sustainable and fundamental characteristic of microbial species assemblages. In addition to dispersal patterns, competition and predation are examples of processes that could allow rare species to thrive for longer periods of time than predicted by neutral dynamics alone[43]. For instance, rarity may not be disadvantageous for a species that is successful in a small, specialized niche, so long as the niche itself is stable over time. Being rare could also protect species from certain predators, which have evolved to capture the dominant prey taxa, yet are not adapted to hunting rare species[43,44]. Such density dependent selection is well documented in many organisms[45] and is seen as one reason why neutral patterns are not found in other groups of studied organisms[46]. Mortality from viral infections, predation pressure, and phytoplankton blooms are additional factors that could serve as population checks and disrupt random walks in radiolarian species abundance[47]. Viruses and predators could maintain low abundances of certain species, whereas phytoplankton blooms may trigger rapid population growth in others. Biotic interactions have been shown to be vital in shaping modern plankton community structure[48], suggesting that such interactions may overwrite neutral population dynamics. Although the specific nature of ecological interactions among the plankton are poorly understood (particularly for extinct taxa), our results are consistent with the idea that they may contribute to extinction patterns over time.

Trophic diversity and habitat variation among radiolarians could also partially explain why the predictions of neutral theory are not supported by our data. Neutral theory in the strict sense is designed to apply to individuals within a single trophic group[1]; however, in practice it is often invoked for more heterogeneous taxonomic groups (e.g., gastropods[14], small mammals[49], passerine birds[50]). Some radiolarians are mixotrophic (species which utilize both autotrophic and heterotrophic strategies), as are many other plankton groups[51], which could complicate the degree to which neutral dynamics should be expected to apply. Radiolarian species are also known to inhabit specific water column depth zones, which creates stacked, ecologically distinct sub-communities of species throughout the world oceans[24,52,53]. Not only does depth control trophic strategy to some extent (e.g., species that host algal symbionts are obligated to dwell in the sunlit ocean), but the growth and decline of populations observed

in the fossil record represents the summed dynamics of an unknown number of sub-communities as well as seasonal turnover within those sub-communities. Another important aspect of radiolarians' vertical habitat distribution is environmental variability. Environmental stability increases with water depth, which could differentially affect the evolutionary rates of taxa based on habitat. Previous work has shown that species living in more stable environments tend to have longer durations than species in more variable environments[54]. Furthermore, cooler temperatures in the deep water masses may contribute to slower metabolic rates in plankton, which in turn affects their rates of mutation and evolutionary turnover[55]. Both environmental stability and cooler temperatures could thus bias deep-dwelling radiolarian species toward greater longevities than their surface-dwelling counterparts, perhaps diluting or negating the impact of population size on extinction risk in our samples. The fossil assemblage in a single sediment sample represents deposition from throughout the water column, and is thus a composite of these sub-communities, which may have varying degrees of distinction and interconnectivity based on the water mass structure of an ocean region at a given time[29]. The assumptions of neutral population dynamics therefore may not easily translate to plankton paleo-communities that include a multitude of distinct assemblages segregated by depth.

Niche theory as an alternative to neutral theory was not supported by our results, which showed no link between higher taxonomic identity or biogeographic range with longevity. Collodaria, Nassellaria, and Spumellaria did not have statistically different species longevities in the EEP or the SO (Fig. 3). Average longevities among these groups were quite different between oceanographic regions, but this is interpreted as an artifact of the time interval lengths of our two datasets, which prevents us from observing any long-ranging species in the EEP but not the SO (see Methods). Our negative result for taxonomically-controlled longevity contrasts with the findings of similar studies on for-aminifera, which concluded that both morphotype and abundance correlated with species average longevity[11,12]. Most collodarians are known to be colonial, whereas nassellarians and spumellarians are solitary[25]. Algal symbionts are found in all described species of Collodaria, but only some species of Nassellaria and Spumellaria, which can also have a wide array of different life strategies, including predatory, herbivorous, and detritus-feeding specialists[56]. Unfortunately, our niche test was limited to these higher taxonomic groupings because not enough is known about individual radiolarian species biology and ecology to test for the influence of specific traits or behaviors on their longevity.

Occupying a large biogeographic range is generally thought to promote the long-term survivorship of a species[57]. Several previous studies on marine macroinvertebrates have supported this idea by demonstrating a negative relationship between range size and extinction risk (e.g., [5,6,58]). However, in other cases a link between these factors is less clear[8]. While the geographic coverage of our SO dataset is broad, the single location used for our tropical dataset limits our ability to fully analyze biogeographic range as a predictor for longevity. Nonetheless we were able to generally categorize each species as either endemic or cosmopolitan based on its occurrence record (see Methods). Results of ANOVA tests and linear mixed effects models indicate that biogeographic range category has no significant influence on radiolarian species longevity (Supplementary Fig. 3). A more geographically comprehensive radiolarian occurrence-abundance dataset that includes range area estimates rather than categorical range assignments could produce more robust results, but a different conclusion from such a study is unlikely given the absence of any pattern in our data and evidence that most low latitude

radiolarian species are pan-tropical[30]. Instead, our findings suggest that ecological interactions among groups[48] and abiotic environmental change[27,59] could be more important factors driving evolution-extinction dynamics in marine plankton.

In the context of contemporary objectives to conserve species diversity, it is common practice to evaluate species risk of endangerment and extinction based largely on their rarity[18]. It is both intuitive and aligned with neutral theory that rare species have a more tenuous existence than abundant species. However, the majority of living species in microbial ecosystems are rare[42] and our results from the fossil record suggest that these rare species may thrive long-term at low abundances. We found no link between species abundance and risk of extinction, challenging the assumption that common species are less likely to go extinct than rare ones. Thus, it is important that we improve our knowledge of other factors contributing to extinction risk, if we wish to make policy decisions that will effectively conserve biodiversity of microorganisms. Our results suggest that factors other than abundance, such as life history traits, feeding strategies, ecological interactions, and tolerance of environmental change could have been primarily responsible for the extinction patterns observed in Neogene plankton. Further investigation into these possible extinction selectivity factors will help us make more accurate risk determinations for marine plankton species, contributing to the overall health of marine ecosystems.

## Methods
**Data and study design.** The samples used for this study are comprised of fossilized polycystine radiolarian specimens obtained from deep sea sediment cores that were processed to include siliceous material in the >45 μm size fraction. Neogene and Quaternary radiolarian abundance counts were made at the species level for Southern Ocean (SO) and eastern equatorial Pacific (EEP) samples and are publicly available[60]. The EEP dataset was first published with Trubovitz et al.[27], while the SO dataset used in this study is a subset of the data collected by Renaudie[61] and summarized in ref. [28]. Our SO selection includes 97 samples from IODP Sites 689, 690, 744, 747, 748, 751, and 1138 (Fig. 1b), all of which have age models rated as "Good" or "Very Good" in the Neptune Sandbox Berlin (NSB) database[62] and show no evidence of age model problems or reworking (such as biostratigraphic marker taxa being abundant outside their well-established ranges). These samples span 22.65–0.04 Ma. The average number of specimens identified to the species level per sample is 7,190: a total of 697,396 species-level observations (Supplementary Data 1). All samples in the SO dataset were enumerated to achieve at least 99% coverage[28]. Coverage is a metric that estimates the percent of biodiversity in an assemblage that has been sampled[63], based on both sample size and assemblage evenness. In addition to observing collection curves (which illustrate the decreasing rate that new taxa are discovered with increasing sample size), coverage provides a way to check that samples have been sufficiently surveyed, and with uniform completeness. The EEP dataset contains 14 samples spanning 10.3–0 Ma, all of which were obtained from the International Ocean Discovery Program (IODP) Site U1337 (Fig. 1c). Approximately 5,200 specimens were identified per sample, with an average of ~2,500 of these specimens identified to the species level: a total of 35,311 species-level observations (see Supplementary Data 1 in ref. [27] and Supplementary Data 1 in this manuscript). Coverage values from the EEP dataset were >90%; values ranged between 93% and 97% (see Supplementary Data 1 in ref. [27]). Species richness and evenness were markedly different between locations. Average within-sample diversity was 356 species (±19) over the last 10.3 my the EEP, but ranged from only ~100–200 species per sample in the SO over the last 22.65 my (Fig. 1a). Due to the differences in study interval length, geographic coverage, sample size, and sample coverage between the EEP and SO datasets, each is analyzed separately to prevent any unnecessary bias that could occur from combining the data.

For each dataset, we picked only the species that had their entire stratigraphic range within the study interval. Any species that had any occurrences in the upper or lower two samples were removed. Then, the average gap size between occurrences was calculated for each species in each dataset, and rounded up to the next whole number. This number constituted the error estimate on each first and last occurrence datum (see Supplementary Data 1; yellow highlighting indicates each species first occurrence and estimated error interval, while red highlighting indicates last occurrence date and estimated error interval). Species that did not have any gaps between occurrences (average gap size = 0) were given the minimum error bar of 1 sample. This technique has long been common practice in biostratigraphy and is similar to Marshall's exploration of estimating confidence intervals on stratigraphic ranges[64,65]. Any species that had its range more than doubled with the addition of the appropriate confidence intervals was removed

from the analysis for having a range too poorly constrained. If a species' longevity error bar extended below or above the study interval, the species in question was removed for either possibly being extant or possibly ranging beyond the beginning of our study interval. A few occurrences were excluded if there was reason to believe they were misidentifications or reworked specimens, based on the ranges of known biostratigraphic marker taxa[66] (these occurrences are marked with red text in Supplementary Data 1). Using this vetting procedure, 189 species in the SO dataset and 101 species in the EEP dataset were determined to have their entire ranges represented in the study intervals.

Longevity was calculated as the midpoint of a species' first occurrence date error bar minus the midpoint of its last occurrence date error bar, giving an estimate of its duration in millions of years (see Supplementary Data 1). After finding the longevity of each species, abundance metrics were calculated. The abundance count data of each species was standardized to relative abundance (%) within each sample, to account for differences in sample size. The relative abundance of each species was calculated per sample across its entire range, as the percent of the entire species-level radiolarian assemblage observed. Genus-level and higher taxon-level observations were removed from this analysis. Stratigraphic plots illustrating the occurrence records of all species-level taxa in these assemblages can be found in Appendix 1 (SO taxa) and Appendix 2 (EEP taxa)[67]. From the relative abundance calculations, we determined the average (mean) relative abundance of each species throughout its range, the maximum relative abundance it ever achieved, and its standard deviation in relative abundance (Supplementary Data 1). Because the raw relative abundance data were highly skewed toward low average and maximum abundances for both SO and EEP datasets, the data were normalized using a natural logarithm transformation to make them suitable for regression analyses.

In a follow-up analysis, species longevities were grouped by higher taxonomic identity (polycystine Orders: Collodaria, Nassellaria, and Spumellaria) to investigate the existence of a taxonomic basis for species longevity. Although it is the consensus among radiolarian workers that Collodaria, Nassellaria, and Spumellaria are considered polycystine Orders[25], molecular phylogenetic analyses indicate that the Collodaria are in fact nested within Nassellaria[68]. For the purposes of this study, the term "Nassellaria" refers to all non-collodarian nassellarians, which have morphological characters separating them from Collodaria. Taxonomic categories of all species are given in Supplementary Data 1 (C = Collodaria, N = Nassellaria, and S = Spumellaria). Collodaria are almost entirely colonial forms, which also makes them ecologically distinct from the exclusively solitary Spumellaria and Nassellaria. Differences in species longevities based on higher taxonomy could be interpreted as evidence of morphological or ecological extinction selectivity and thus support niche theory rather than neutral theory. Radiolarian species' average longevities are not well established, but prior studies suggest median values between ~5 and 10 my[31,35], with a negatively skewed distribution (i.e., a long tail of long ranging taxa). Mean longevity estimates for radiolarian species in previous literature vary from 4.8[31] to 12.9 my[35]. The relatively short length of the EEP study interval (10.3 my) thus likely does not capture any of these longer-ranging species, and is biased toward representing the short and average ranging taxa. The longer length of time included in the SO dataset enables it to reflect a fuller range of taxon longevities that includes many of the longer-ranging taxa. Therefore, we cannot directly compare the longevities in the EEP with the SO, but each dataset can be interpreted as an independent test of taxonomic control over radiolarian extinction risk.

Another potential predictor of longevity we considered was species geographic range breadth. Some paleontological studies have found that occupying a large geographic range lowers a taxon's susceptibility to extinction[5–7], while others found that range size is not a strong predictor of extinction risk[8]. To explore this pattern for radiolarians, we began by classifying each species as either endemic or cosmopolitan based on whether it was observed in only the EEP or SO, or both regions (in Supplementary Data 1: E = endemic and C = cosmopolitan). Species lists and photographs of undescribed taxa were carefully cross-checked between regions. Although we lack midlatitude species occurrence data and the biogeographic variables in our dataset are coarsely categorical rather than continuous values, our dataset still allows us to broadly characterize whether species longevity is related to biogeographic range size.

Because microfossils are so small and abundant, only a tiny amount of sediment (~1 cc) was processed per sample, yielding tens of thousands of radiolarian specimens per sample at our study sites. This gives us a snapshot in time (on the order of ~5–10 thousand years, depending on sedimentation rate and local bioturbation intensity)[69]. This allows us to collect data with higher fidelity to the assemblage structure of real paleocommunities compared with the large time-binned data often used in macroinvertebrate paleontology syntheses, which are on the order of at least millions of years, but sometimes >10 million years (i.e.,[8,14]). Although paleocommunities will almost always have some degree of time-averaging (an assemblage of taxa that were not strictly coeval), our samples are probably closer to capturing true relative abundances in past assemblages than most macroinvertebrate datasets of comparable sample size, because there is a limited amount of time-averaging in small (by volume) deep sea sediment samples compared with integrated data from shallow marine bedding surfaces. Like other paleontological datasets, ours is also subject to biases from differential preservation potential among taxa. All polycystine radiolarian skeletons are built of opaline silica, but some are more robust than others and may be more likely to get preserved in the fossil record. It is also unclear the extent to which niche position in

the water column affects preservation potential. For instance, deeper-dwelling species may have a preservational advantage because they have a shorter distance to travel to reach the sediment-water interface, limiting the chances of skeletal breakage or dissolution. Studies have found that most radiolarian diversity (>90%) in living assemblages does successfully get preserved in the fossil record[24,26]. The impacts of preservation bias on relative abundance are poorly known, but the relative abundance rankings for most species seem to be broadly consistent between the water column and the surface sediment, at least in well-preserved samples[70]. However, comparisons between the water column and surface sediments are complicated by the fact that sampling is on very different time scales (usually <1 year for the water column versus a sediment composite of several thousand years).

While it is nearly universal practice to measure community composition from relative data in paleoecological analyses, there is an inherent risk that relative abundance between samples may not always accurately reflect true changes in species population sizes. This can occur if sample size and sediment accumulation rate varied considerably from sample to sample. Almost uniquely among paleoecological datasets, marine microfossil data from deep sea drill cores have well-known sediment accumulation rates[62]; thus, we were able to control for a potential inversion between relative and absolute abundance in two ways. First, we allowed only slight variations in sample size within the EEP and SO datasets (Supplementary Data 1). Second, the EEP dataset is comprised of data from a single IODP Site (U1337), which experienced near constant rates of sedimentation throughout the last 10 my (see ref.[27] and references therein). Therefore, we expect any deviation from a direct relationship between relative abundance and absolute abundance to be very minor (ca. a factor of 2 or less), and not significantly affect the results or interpretations we present here, which are based on variations of three to four orders of magnitude in relative abundance.

**Statistics and Reproducibility**. Relative abundance values were transformed by natural logarithm to normalize the data prior to regression analyses. Four separate single linear regression analyses were performed on the SO and EEP datasets, to determine whether there is a correlation between either mean or maximum relative abundance (independent variable) and longevity (dependent variable). Results were assessed based on a 95% confidence interval ($p = 0.05$) and R-squared values of each regression. Log transformation and linear regression analyses were performed in R v. 3.6.0[71]. Plots were made using built-in R functions as well as the R package ggplot2[72] (Fig. 2). Standard deviation in abundance was also considered as a potential predictor for longevity, but like the other regression analyses it produced nonsignificant results (Supplementary Fig. 2).

An analysis of variance (ANOVA) was conducted for the EEP and SO datasets separately to test whether the longevities of the polycystine Orders Collodaria, Nassellaria, and Spumellaria were statistically different from each other. Visual inspection of the median and range of longevities was made using violin and box plots (Fig. 3). This analysis was performed using functions built into R v. 3.6.0[71] and illustrated with ggplot2[72]. Similarly, ANOVA tests were used to determine whether the longevities of species classified as cosmopolitan versus endemic were significantly different from each other. This analysis was also performed separately on SO and EEP species to avoid introducing bias due to structural differences in the regional datasets.

Linear mixed effects models using biogeographic category and average relative abundance as the combined independent variable and longevity as the dependent variable were built using the R package lme4[73], and compared with the simple linear regression models of relative abundance versus longevity based on Akaike Information Criterion values ("AIC" function in R v. 3.6.0[71]). The "boxplot" function in R v. 3.6.0[71] and the package ggplot2[72] were used for visualization (Supplementary Fig. 3).

**Reporting summary**. Further information on research design is available in the Nature Portfolio Reporting Summary linked to this article.

## Data availability

All data used in this study are presented in Supplementary Data 1, which is a subset previously published data[27,28,60] available for download from the Zenodo repository: https://doi.org/10.5281/zenodo.4014322. Additional data that support the findings of this study have been deposited in Figshare: https://doi.org/10.6084/m9.figshare.22637566.v1.

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

## Acknowledgements

This work was supported by the U.S Science Support Program Schlanger Fellowship (S.T.), the Geological Society of America Graduate Student Research Grant (S.T.), the Paleontological Society Student Research Grant (S.T.), the Cushman Foundation for Foraminiferal Research Loeblich and Tappan Student Research Award (S.T.), the German Academic Exchange Service (DAAD) student grant (S.T.), and The Federal Ministry of Education and Research (BMBF) under the "Make our Planet Great Again" German Research Initiative, grant number 57429681, implemented by the German Academic Exchange Service (DAAD) (J.R., D.L.). We also thank the International Ocean Discovery Program (IODP) for providing sediment samples and Fr. Sylvia Dietze (Museum für Naturkunde) for sample preparation.

## Author contributions

Conceptualization: D.L., S.T., J.R., P.J.N. Methodology: S.T., D.L. Investigation: S.T., J.R. Visualization: S.T. Supervision: D.L., J.R., P.J.N. Writing—original draft: S.T. Writing—review & editing: D.L., J.R., S.T., P.J.N.

## Funding

## Competing interests

The authors declare no competing interests.
