## [Peer Review File · Communications Biology]

Reviewers' comments:

Reviewer #1 (Remarks to the Author):

The manuscript by Trubovitz et al. entitled "Abundance does not predict extinction risk in the fossil record of marine plankton" presents an analysis of the likelihood of extinction of Radiolarian in the fossil record in two markedly distinct environments. The main conclusion of the work (contained in the title) is that the longevity of radiolarian species in the fossil record is not linked to their relative abundance, in other words, abundance does not protect species from extinction.

I must say as a disclaimer that my speciality is centred on modern foraminifera ecology and diversity. Therefore, the remarks I will make in this review might be naïve but they may serve to the author as a point of reference of the perception of their work by a non-specialized whom as a keen interest on macro ecological analyses in the fossil record, which is my case.

I enjoyed reading the paper although I found it too long. The analysis is based on a strong dataset that has been counted by the authors of the study, which ensure consistency of the analysis. Although the authors benefit of this dataset, I find that the analyses they have done, and the way the present their data is slightly unrefined. For instance, the authors could show a map to provide the core location (especially because they use a composite record), a time scale and represent the variation of diversity, evenness through time. I understand that the dataset has already been presented elsewhere (<https://www.nature.com/articles/s41467-020-18879-7>) and that they may not want to produce the exact same graphs, but the present manuscript would benefit to have at least some elements of the Fig. 2 of their paper in Nature communication to provide background information. I think figures 1 and 2, and figures 3 and 4 should be merged as 1 as they show the same information and having them next to each other will ease the comparison between SO and EEP, and I feel that the Table 1 is not really needed.

I have four major comments regarding the manuscript:

- In the fossil record, we always analyse relative abundance that is a measure of relative reproductive success between species. Hence, we never access the true abundance of species that can vary drastically depending of season and geographic range. I have not seen in the paper any information on the reproduction strategy of radiolarians and the effect of seasonality, specimens longevity, life cycle etc... on the production of the fossil record. I think it will be useful to have such information because fossil assemblages are not perfect pictures of paleo-ecosystems. For instance in foraminifera, fossil assemblages are sometimes dominated by species which are extremely abundant during seasonal blooms and quasi absent the rest of the year. Could the authors provide more about these points in the introduction? This is not a critic but rather a clarification for the audience not working with fossil assemblages that relative abundance is an imperfect measure of the occupancy of the ecological niche by the fossil species.

- The definition of fossil species concept for radiolarian is in my opinion missing. Again, I am biased by my view of evolution of planktonic foraminifera where it exists 2 extreme situation: The cladogenesis and the anagenesis. In the first case, one can clearly identify the stratigraphic range of a species but in the second, there is no clear beginning nor end to a taxa range, only arbitrary limits defined by the taxonomist. I would like to know if there is a lot of anagenetic lineages in the fossil record of radiolarian as it may influence the living range of the species.

- Given the conclusion of the author, I was surprised to not find any reference to the red queen theory of Van Valen. I think that beyond single taxa survivorship, the author could add another analysis and show the survivor ship of the ecosystem between the SO and EEP by showing a Linear survivorship curves (See fig 1 in: (<https://www.sciencedirect.com/science/article/pii/S0169534711000863>)). This could be informative to discuss how long it takes for the complete turnover of ecosystems between the two environments.

- I think that there is one point missing in the paper and it is the concept of trophic species. Individual taxa may appear and disappear through time but this has no effect on the ecosystem as long as the trophic role they occupy is maintained. I do not know if there is way to assume which role radiolarian species assumed in the past, especially when a large part may have been mixotrophic but I feel that the manuscript will gain in discussing whether or not the same ecological role are maintained through time or not.

Overall I think the paper length should be greatly reduce (By at least a 1/3, more if possible) to have a terse paper of 3-4 pages. I think it will benefit the manuscript and increase its visibility.

Detailed comments.

Abstract:

I think the author should mention only relative abundance in the title and throughout the text and in the title.

Introduction.

I feel that the introduction is too long and gives too much light to the work on other fossil groups such as bivalves. The intro from Lines 31-57 is straightforward, Lines 58-75 is odd because it explains the hypothesis that will be tested but feels detached from radiolarian fossil record (I would move this more in the methods), Lines 76-92 puts too much emphasis on other fossil groups (I would move this more in the discussion). Then a large segment of the intro (Line 93-126) is dedicated to results on planktonic foraminifera but cites 2 work that are 30 years old and comment then extensively the work of Allen and Savage. Again, this feels more like discussion and it really distract from the manuscript of the authors. Then Lines 127-153 is back to the radiolarians. I feel that the ellipse from lines 58-127 is not needed and I would replace it with information about basic biology of radiolarians and the specificity of their fossil record instead.

Results.

I think the transition between Introduction and Results could be improved. When the Method section comes at the end of the manuscript, the reader should be provided with a simplified explanation of what was done to help to interpret the result without having to go back and forth between the Result and Method sections. As specified above, I think the authors should provide one "contextual figure to show the geographic location of the core, the stratigraphic coverage, variation of diversity through time etc... to have an understanding of the context and not only plots with correlations which is too dry.

Line 167-168. I do not know what the authors mean by "Visual inspection of the data".

Line 169-173. Writing the figure legend in the main text is not needed. Please remove.

Lines 180-182. Same.

Lines 200-203. This feels like discussion.

Lines 183-214. Although it is unlikely to change the outcome, I wonder if the author could do a two way ANOVA where they test together for the two factors "Location" and "Morphogroup" as well as the interaction between the factors. I think it is better than doing separate tests.

I have a general critic regarding the choice to do the analyses on average and maximum abundance. This are two oversimplifying variable that do not represent the actual variability of the dataset. Could the author provide an additional correlation with SD too?

Discussion

Lines 235-238. Figures is needed to support the sentence.

Lines 240-242. Same.

Lines 243-290. I respect the choice of the author to give an overview of other results from the literature but I am somehow bothered by the fact these fossil group occupy different environment and have different life strategies too. I am not certain that here the comparison is relevant, beyond the differences in the dataset structure mentioned by the authors.

Lines 291-315. I would reduce this entire section to a few sentences. It is well known that ecological theories are often formulated based on observation made in terrestrial environment and are hence hard to transpose on marine environments. I think the authors spend too much time on explaining why such model do not fit, and they should what their data means, which is more interesting in my opinion.

Lines 316-350. Same, although there at least more element on radiolarians in that section.

Lines 351-370. This part should be reduced as well.

Lines 382-385. These studies are 30 years old. I do not question the validity of works done in the past, I am only surprised that the authors do not mentioned more recent work such as Ezard et al. (2011; <https://www.science.org/doi/full/10.1126/science.1203060>), Rillo et al. (2019 ; <https://onlinelibrary.wiley.com/doi/full/10.1111/geb.13000>), Antell et al. (2021 ; <https://www.pnas.org/doi/10.1073/pnas.2017105118>). Although not necessarily on the spot of what the manuscript is presenting, I am certain that it exists more recent literature worth mentioning for the discussion.

Lines 401-403. Arriving at this stage of the paper, I wonder why the author did not give any light

on the rare biosphere from a microbial perspective (<https://www.nature.com/articles/ismej2016174>). I feel that the paper is heavily dominated by ideas derived from macro organisms and the author do not consider enough the existing literature is protistology.

Lines 407-415. I think the authors have missed one important reference with the work of Lima-Mendez (2015; <https://www.science.org/doi/full/10.1126/science.1262073>). Trophic interactions is the main factor structuring ecosystems far ahead from abiotic factors. I think this is worth mentioning this fact at the end of the article that the rise and fall of species through time has taken place into that context. Biotic factor have been mentioned in the discussion but I think that they have not been put enough forward (But I am certainly biased).

Methods.

I have no major critics on the methods since the dataset has been reused, except that this section as the rest of the paper is too long and wordy. I suggest reducing it to stick to the essential.

I wish success to the authors for the future work on the manuscript.

Sincerely yours,
Raphael Morard

Reviewer #2 (Remarks to the Author):

Trubovitz et al use the radiolarian fossil record to test for the relationship between the average (and maximum) relative abundance of a species (throughout its existence) and its longevity across the Neogene and Quaternary periods. Probing this relationship is a timely, very relevant scientific question, as many conservation efforts are decided based on species' abundance estimates. The authors make the important connection between palaeoecology and conservation biology clear in the introduction. The manuscript is well written, and the fossil dataset used is taxonomically consistent to the species level (rare in palaeoecology), with many specimens sampled per sample and very good coverage. The authors use linear regression to show that the expected positive relationship between abundance and species' persistence is not present in the radiolarian fossil record of two regions (Southern Ocean and Eastern Equatorial Pacific). The absence of a relationship between extinction risk and population abundance is intriguing, although I have some remarks about the statistical analyses and the spatial coverage of the study (see below).

[1] The authors' expectation of a positive relationship between abundance and longevity is based on the neutral theory of biodiversity (well introduced in the manuscript). The neutral theory is based on a stochastic demographic model (ie, all individuals regardless of species identity have same chances of dying and colonizing/giving birth). When most species are rare (which seems to be the case as relative abundances of radiolarian were "highly skewed towards low values" - lines 481-484), I would expect the relationship between abundance and longevity to be weak because there is variance around the stochastic process of death ("random walk into extinction", line 69). Thus, the difference in longevity between two species with small number individuals probably falls within this variance and is not expected to show a statistically significant difference. One way to estimate this variance would be to have a null, neutral demographic model that would constrain the expected relationship between species longevity and abundance. I understand that the computational simulation of a null model is beyond the scope of this empirically-oriented manuscript, but the variance of the stochastic process might lead to a non-linear pattern between abundance and longevity of no relationship in the low-abundance spectrum and positive relationship once species become more abundant. Moreover, given that former studies have found non-linear relationship between abundance and extinction rates (Simpson & Harnik 2009 Paleobiology [Ref. 13]: "rare and abundant genera exhibiting rates elevated over those of genera of moderate abundance"), I would expect this U-shaped pattern to also be tested with the radiolarian dataset and the final model be selected using a model selection approach (instead of visually from the plots - line 230).

[2] Linear regressions have the underlying assumption of independent observations. The fossil dataset used, however, potentially break this assumption because, for example, the taxa are phylogenetically related, and/or the data points come from a time series that might have some type of temporal autocorrelation (eg, changes in sedimentation rate along the core). Ideally, these dependencies would be taken into account (eg, linear mixed effect model) or the authors would show that they do not affect the results (eg, there is no phylogenetic signal). Also, providing the residual plots of statistical models in the Supp Information is recommended to show that the data does not break model assumptions.

[3] Former work by Harnik (Ref 5) show that geographical range, not abundance, is the main predictor of species extinction risk. Therefore, I expected the authors to discuss the geographical range of species more thoroughly in their paper. In lines 142-143 they state: "[our study] includes reasonably complete sampling of species ranges given what we know of their biogeographic distribution patterns." However, my understanding from the methods is that the SO region was sampled in 7 locations (IODP sites) and EEP in one single site, so it is not clear to me how this spatial coverage is enough to comprise the geographical ranges of the hundreds of studied species. Without knowing how endemic or global are these species, it is difficult to know to what degree does extinction in the IODP sites reflect global extinction of a given species. Or, even, if the IODP sites are on the edges of a species' range, its relative abundance might not be representative of its overall relative abundance (ie, across its range).

[4] Since the authors focus solely on abundance as an explanatory variable for species persistence, I expected them to have a more in-depth analysis of this variable. For example, the authors use average and maximum abundance of a species throughout its existence; however, how does the abundance of each species vary throughout its existence? How representative is the max and mean abundance of a species' across its existence? Do species decrease in abundance before they go extinct? How does the species abundance distribution (SAD) of the assemblage in each region looks like? The shape/skewness of the SAD informs us about the relative proportion of rare vs abundant species, giving insights of the expectation of the abundance-longevity relationship for each region. Does the SAD change through time? Ie, was there a moment when many rare species or many abundant species went extinct? Also, the true maximum can be sensitive to outliers, did you consider using the 90th or 95th percentile of the data instead? "Average" can sometimes be interpreted as mean, median, mode – I suggest being precise about which summary statistic you calculated.

Table 1 and Figures 1 and 2 overlap in information. You could choose to remove the table (adding the significance information visually to the figure with dashed vs. solid lines) and show new figures that relate to a more in-depth analyses of species abundances, as suggested above. Also, Figure 3bcd and 4bcd seem to repeat information in Figure 5 (vice-versa). You could choose to show instead of boxplot (Fig 5), a violin plot where one can see more clearly the variation in the data including more info that is in the barplots (Fig 3bcd and 4bcd).

Introduction

Introduction is well written, and this comment is merely a style preference: lines 50-57 are in my opinion a bit early and too detailed for the first paragraph of the intro. I would keep the research question more general here and then come back to the details of data and specific statistical tests at the end of the intro (in the paragraph that starts at line 127).

Line 58: this paragraph starts a bit disconnected from the one before.

Line 59-60: add reference to this sentence.

Line 127: I would remove "Given the inconclusive results of previous work," - it is not completely true and, in my opinion, not necessary as the reasoning and relevance of your work is very clear already.

Lines 157-160: It is good practise to provide the residuals plots of the models to show that statistical assumptions such as independence of observations are met

Methods

Line 541: affects

Paragraph 552: I think the discussion of relative vs. absolute abundance (or parts of it) should be in the discussion section not in the methods because it is a very important difference between contemporary and palaeo ecological data. The expectation of high abundance related to low extinction rates in ecological theory is based on absolute (not relative) abundance. So, one of the reasons you did not find a relationship could be because median/maximum relative abundance of a species is not a good proxy for its median/maximum absolute abundance.

Lines 558-560: Is there a reference to this statement? And I am not sure I understood the rationale of these statements here... even if sedimentation rates are very well constrained/known, shouldn't they be considered then when calculating abundance in the sediment to try to get to a better estimate of absolute abundance? Wouldn't the way to control for an inversion of absolute vs. relative abundance be to keep sampled mass constant (instead of number of individuals)?

Lines 565-569: Still in the same rationale paragraph, it is not clear to me why do you think the inversion would be ca. by a factor of 2 or less.

Discussion

Line 217-219: I would say "the theory *predicts* a *negative* relationship between abundance and extinction risk". The theory itself is a metacommunity demographic model, where each individual has the same chance of dying regardless from which species it is from. So, it predicts that species with lower abundances have a higher chance of going extinct because it has less individuals to persist during the stochastic demographics of the model.

Line 244: Harnik PNAS 2011 actually shows a positive relationship between abundance and species duration. This relationship is weakened when he takes the interaction between abundance and geographical range into account. Because these two variables correlate, and range is a stronger predictor, then abundance becomes a weak predictor. Because you did not consider geographical range, I would not say your results agree with Harnik's result.

Line 245-246: "evolutionary behaviour" is a strange term. Prefer "evolutionary processes"

Line 248: if extinction is not neutral, can you say something about common traits of species that went extinct or a deterministic process that selectively drove these species to extinction?

Line 273 onwards: When discussing the contrasting results between your work with those focussed on marine macroinvertebrates, it would be interesting to hear the authors' opinion about life history differences between these groups (instead of only data structure differences). Could it be that the greater dispersal capability of plankton (radiolarian) makes them more resilient to extinction than bivalve/brachiopod even when a species has low abundances? In this sense, discussing geographic range and shifts in geographic ranges of the studied species is relevant.

Line 291: re-phrase this sentence as you did not test how your results compare to a null model generated under the neutral assumption.

Line 324: remove 'important'

Line 347-350: could you use your data to test this hypothesis? Do you know which species are deep vs surface-dwellers?

Line 352: remove comma between subject and verb

Line 366: "they [ecological interactions] do indeed exist" sounds strange plus this was not explicitly tested to say they "contribute significantly". I would re- write: "but the present study supports the idea that they [ecological interactions] may affect extinction dynamics over time."

Line 371: Remove "(at least at the level of higher taxa)" – it is confusing here, and it is made clear in line 373

Line 393: 'among' instead of 'between'? It seems there is also a lot of variation in biology and ecology *within* higher taxa, is this true? If so, this variation could explain why there was no clear difference among these higher taxa.

Line 410-412: This sentence should be more speculative since you did not test for life history traits, feeding strategies, interactions, tolerance. Something like: "Our results suggest that factors other than abundance, such as [...], *could have been* responsible for the extinction patterns observed in Neogene plankton."

Reference list

Re-check all reference list, as there seem to be duplicates (eg, 5 & 24, 13 & 22).

Supplementary tables

Regarding the supplementary table: what do the colours mean? Some cells are coloured yellow or rose

Signed,

Marina C. Rillo

Reviewer #1

Opening Remarks

Comment: The manuscript by Trubovitz et al. entitled “Abundance does not predict extinction risk in the fossil record of marine plankton” presents an analysis of the likelihood of extinction of Radiolarian in the fossil record in two markedly distinct environments. The main conclusion of the work (contained in the title) is that the longevity of radiolarian species in the fossil record is not linked to their relative abundance, in other words, abundance does not protect species from extinction. I must say as a disclaimer that my specialty is centered on modern foraminifera ecology and diversity. Therefore, the remarks I will make in this review might be naïve but they may serve to the author as a point of reference of the perception of their work by a non-specialized whom as a keen interest on macro ecological analyses in the fossil record, which is my case.

I enjoyed reading the paper although I found it too long. The analysis is based on a strong dataset that has been counted by the authors of the study, which ensure consistency of the analysis. Although the authors benefit of this dataset, I find that the analyses they have done, and the way the present their data is slightly unrefined. For instance, the authors could show a map to provide the core location (especially because they use a composite record), a time scale and represent the variation of diversity, evenness through time. I understand that the dataset has already been presented elsewhere (<https://www.nature.com/articles/s41467-020-18879-7>) and that they may not want to produce the exact same graphs, but the present manuscript would benefit to have at least some elements of the Fig. 2 of their paper in Nature communication to provide background information. I think figures 1 and 2, and figures 3 and 4 should be merged as 1 as they show the same information and having them next to each other will ease the comparison between SO and EEP, and I feel that the Table 1 is not really needed.

Response: We appreciate the suggestion and have made major changes to our figures to address it. As requested, we have added a new figure with background information, Figure 1, which illustrates diversity trends through time in addition to sample localities for the EEP and SO sites used in this study. Also as requested, we have now combined the original Figures 1 and 2 into a single figure - Figure 2 in the revised manuscript. The information presented in Table 1 in the initial manuscript has been integrated with Figure 2 in the revised manuscript, to reduce redundancy. We also merged Figures 3 and 4 together with Figure 5, which is now Figure 3 in the revised manuscript.

Major Comments

Comment: In the fossil record, we always analyse relative abundance that is a measure of relative reproductive success between species. Hence, we never access the true abundance of species that can vary drastically depending of season and geographic range. I have not seen in the paper any information on the reproduction strategy of radiolarians and the effect of seasonality, specimens longevity, life cycle etc... on the production of the fossil record. I think it will be useful to have such information because fossil assemblages are not perfect pictures of paleo-ecosystems. For instance in foraminifera, fossil assemblages are sometimes dominated by species which are extremely abundant during seasonal blooms and quasi absent the rest of the year. Could the authors provide more about these points in the introduction? This is not a

critic but rather a clarification for the audience not working with fossil assemblages that relative abundance is an imperfect measure of the occupancy of the ecological niche by the fossil species.

Response: We address the issue of fossil assemblage biases and how they relate to radiolarian data in the last paragraph of the "Data & study design" subsection of "Methods". Unfortunately, not much is known about modern radiolarian life cycles, reproduction, or seasonal succession. Although their paleocommunities are certainly not assumed to be perfect representations of past communities, the few studies exploring this issue have found surprisingly good agreement between living assemblages and surface sediment samples in terms of species richness and relative abundance (cited in the manuscript text: Lazarus 2005, Boltovskoy 1993). Because we are hoping to keep the Introduction succinct and interesting to a broad audience of both biologists and paleontologists, we have chosen to put this information in the Methods section rather than the Introduction.

Comment: The definition of fossil species concept for radiolarian is in my opinion missing. Again, I am biased by my view of evolution of planktonic foraminifera where it exists 2 extreme situation: The cladogenesis and the anagenesis. In the first case, one can clearly identify the stratigraphic range of a species but in the second, there is no clear beginning nor end to a taxa range, only arbitrary limits defined by the taxonomist. I would like to know if there is a lot of anagenetic lineages in the fossil record of radiolarian as it may influence the living range of the species.

Response: There are some anagenetic lineages in Radiolaria (e.g., Sanfilippo and Riedel 1992; <https://doi.org/10.2307/1485841>), and others that evolve via both cladogenesis and gradual change (Lazarus 1986; <https://doi.org/10.1017/S0094837300013646>). The morphospecies definitions we used in this study are based on described species in the literature, and the experience of the authors in examining tens of thousands of specimens. While in some cases the acceptable range of morphologic variability within a species is subjective, radiolarian species concepts are routinely used for biostratigraphy of marine sediments (i.e., Neptune database), so they are generally reliable. In addition, where arbitrary or subjective species limits exist, we would consider these to be randomly distributed errors, not a source of systematic bias affecting the temporal range of particular groups of taxa.

Comment: Given the conclusion of the author, I was surprised to not find any reference to the red queen theory of Van Valen. I think that beyond single taxa survivorship, the author could add another analysis and show the survivorship of the ecosystem between the SO and EEP by showing a Linear survivorship curves (See fig 1 in: (<https://www.sciencedirect.com/science/article/pii/S0169534711000863>)). This could be informative to discuss how long it takes for the complete turnover of ecosystems between the two environments.

Response: While the Red Queen Hypothesis might be an interesting addition to the manuscript, we do not consider it to be vital in understanding our main results. In the interest of keeping this manuscript as focused as possible, we have not included a discussion of the Red Queen Hypothesis. We feel that the addition of linear survivorship curves would be partially redundant with Figure 3, because this figure already displays longevity distributions in each environment. Furthermore, as noted in the manuscript, we think that a direct comparison between the SO and EEP datasets would be misleading, given the differences in study interval

and frequency of samples throughout those intervals. To avoid introducing unnecessary bias, we are keeping analyses of the two datasets independent of one another. We hope, however, that generating a longer and more complete time series dataset from the EEP as part of future research will allow us to quantitatively compare evolutionary dynamics in the two regions. Unfortunately we just can't do that with our existing dataset.

Comment: I think that there is one point missing in the paper and it is the concept of trophic species. Individual taxa may appear and disappear through time but this has no effect on the ecosystem as long as the trophic role they occupy is maintained. I do not know if there is way to assume which role radiolarian species assumed in the past, especially when a large part may have been mixotrophic but I feel that the manuscript will gain in discussing whether or not the same ecological role are maintained through time or not.

Response: This is an excellent point, and we agree it would be very interesting to determine whether the same ecological roles were maintained by different species assemblages over time. However, as mentioned in the "Discussion" section, the specific ecological roles of most living radiolarian species are unknown. Thus, it would be very difficult and speculative to extrapolate the trophic roles of the extinct species in our datasets. While the question of radiolarian trophic roles through time is certainly an intriguing one, we regret that we are still unable to address this given the status of current knowledge.

Comment: Overall I think the paper length should be greatly reduce (By at least a 1/3, more if possible) to have a terse paper of 3-4 pages. I think it will benefit the manuscript and increase its visibility.

Response: We have made an effort to streamline the manuscript wherever possible. Part of Introduction has been shortened and moved to the Discussion. We also removed the "rationale" section from the Methods. Any instances of redundancy we found were also removed. Many of our figures in the original manuscript were changed and combined for concision. However, in order to address all reviewers' concerns, the overall length of the manuscript is on par with its original length. This is mainly due to the fact that we added a new analysis of biogeography, and a few new considerations to the discussion section. We hope that these changes provide readers with a better understanding of the significance of our results without being unnecessarily long.

Detailed Comments

Abstract

Comment: I think the author should mention only relative abundance in the title and throughout the text and in the title.

Response: We refer to "relative abundance" throughout the text and explain how this metric was calculated in our "Methods" section. In order to keep the title as concise as possible, we have simplified the term to just "abundance."

Introduction

Comment: I feel that the introduction is too long and gives too much light to the work on other fossil groups such as bivalves. The intro from Lines 31-57 is straightforward, Lines 58-75 is odd because it explains the hypothesis that will be tested but feels detached from radiolarian fossil record (I would move this more in the methods), Lines 76-92 puts too much emphasis on other fossil groups (I would move this more in the discussion). Then a large segment of the intro (Line 93-126) is dedicated to results on planktonic foraminifera but cites 2 work that are 30 years old and comment then extensively the work of Allen and Savage. Again, this feels more like discussion and it really distract from the manuscript of the authors. Then Lines 127-153 is back to the radiolarians. I feel that the ellipse from lines 58-127 is not needed and I would replace it with information about basic biology of radiolarians and the specificity of their fossil record instead.

Response: We have now added a transition sentence between the first two paragraphs of the Introduction so that it reads less disjointed. We do feel, however, that it is critical to explain the theory behind the hypothesis we are testing early in the manuscript. Otherwise, it might be difficult for some readers to understand the context of our work and it may be less appealing to a broad audience. However, we agree with the suggestion to move lines 76-126 to the Discussion. We think moving this information on other fossil groups greatly helps streamline the manuscript. At the reviewer's request, we have also added a brief description of radiolarian biology and ecology (lines 75-79 in revised manuscript). Because neutral theory is not dependent on niche or biological traits, and few details are known about radiolarian biology, we have chosen to keep this section short.

Comment: I think the transition between Introduction and Results could be improved. When the Method section comes at the end of the manuscript, the reader should be provided with a simplified explanation of what was done to help to interpret the result without having to go back and forth between the Result and Method sections. As specified above, I think the authors should provide one "contextual figure to show the geographic location of the core, the stratigraphic coverage, variation of diversity through time etc... to have an understanding of the context and not only plots with correlations which is too dry.

Response: Our revised manuscript includes a new, smoother transition between the Introduction and Discussion sections. We have also added a contextual figure showing the core locations and variation in diversity through time (Figure 1 in revised manuscript). We already include a simplified description of our methods in the Introduction, so we think it would be too repetitive to also describe the study design in the Results section.

Comment: Line 167-168. I do not know what the authors mean by "Visual inspection of the data".

Response: Phrase changed to "visual assessment" (line 127).

Comment: Line 169-173. Writing the figure legend in the main text is not needed. Please remove.

Response: Lines 169-173 have been removed.

Comment: Lines 180-182. Same.

Response: Lines removed.

Comment: Lines 200-203. This feels like discussion.

Response: We feel this statement should come when the results are first introduced, because the reader will probably wonder why there is a major difference in longevity values between the EEP and SO. We want to explain this point before moving on with the results section, because otherwise the meaning of our findings within each regional dataset may not be fully understood. (Lines 148-149 and 163-164).

Comment: Lines 183-214. Although it is unlikely to change the outcome, I wonder if the author could do a two way ANOVA where they test together for the two factors "Location" and "Morphogroup" as well as the interaction between the factors. I think it is better than doing separate tests.

Response: A two-way ANOVA would indeed be a good choice if the SO and EEP datasets were comparable in temporal length and frequency of sample intervals. However, since the SO dataset includes more frequent samples and covers the last ~22 million years whereas the EEP dataset only covers the last ~10 million years, we will find an artificial correlation between "location" and "longevity" that will impact the results of a 2-way ANOVA. Species in the SO will have greater longevity than EEP species on average, because the SO dataset is able to include long-ranging species, not because SO species actually persisted longer. To avoid misleading results, we have thus analyzed each dataset independently.

Comment: I have a general critic regarding the choice to do the analyses on average and maximum abundance. This are two oversimplifying variable that do not represent the actual variability of the dataset. Could the author provide an additional correlation with SD too?

Response: This is a good point that our metrics may be oversimplifying. The revised manuscript now includes a regression analysis of standard deviation versus longevity (Supplementary Figure 2). Like the other analyses in our study, we found no significant correlation between variables. For interested readers, we have also added a standard deviation in abundance column for each species in Supplementary Tables 1 & 2.

Discussion

Comment: Lines 235-238. Figures is needed to support the sentence.

Response: Figure 1 was added to support this.

Comment: Lines 240-242. Same.

Response: We feel that a full treatment of extinction rate changes in the SO is outside the scope of this study, and would detract from the main points we hope to present here. In addition, we discuss this subject in detail in Trubovitz et al. (2020) which is cited in this manuscript, so interested readers will be able to easily find that information if they choose. We agree that this topic is related to our present study, but it is not fundamental to understanding our results.

Comment: Lines 243-290. I respect the choice of the author to give an overview of other results from the literature but I am somehow bothered by the fact these fossil group occupy different environment and have different life strategies too. I am not certain that here the comparison is relevant, beyond the differences in the dataset structure mentioned by the authors.

Response: According to neutral theory, environment and life strategy should not influence the extent to which the theory applies, provided that individual trophic groups are treated separately. Therefore, we believe a comparison between our study and previous work on macroinvertebrates is valid, particularly since there are few similar studies on plankton. We are precisely trying to show, however, that neutral theory appears to poorly fit the record of plankton dynamics. In the Discussion section we detail some of the reasons this may be the case, including the different dispersal capabilities of plankton versus larger organisms.

Comment: Lines 291-315. I would reduce this entire section to a few sentences. It is well known that ecological theories are often formulated based on observation made in terrestrial environment and are hence hard to transpose on marine environments. I think the authors spend too much time on explaining why such model do not fit, and they should what their data means, which is more interesting in my opinion.

Response: We have rewritten this section to reduce the original content on marine versus terrestrial environments, and add discussions of other factors raised by both reviewers (Discussion beginning at line 329).

Comment: Lines 316-350. Same, although there at least more element on radiolarians in that section.

Response: We believe this section provides important context for the reader to understand the meaning of our results, and have thus kept most of this content but streamlined where possible.

Comment: Lines 351-370. This part should be reduced as well.

Response: We combined this section with the paragraph above it.

Comment: Lines 382-385. These studies are 30 years old. I do not question the validity of works done in the past, I am only surprised that the authors do not mentioned more recent work such as Ezard et al. (2011; <https://www.science.org/doi/full/10.1126/science.1203060>), Rillo et al. (2019 ; <https://onlinelibrary.wiley.com/doi/full/10.1111/geb.13000>), Antell et al. (2021 ; <https://www.pnas.org/doi/10.1073/pnas.2017105118>). Although not necessarily on the spot of what the manuscript is presenting, I am certain that it exists more recent literature worth mentioning for the discussion.

Response: While these are interesting and related studies, we do not see these as directly relevant to the point being made in lines 382-385. We have searched for more recent literature on factors influencing plankton species longevity, but were unable to find anything as relevant as the sources already cited (references 11 and 12). We have, however, added Ezard et al. (2011) as a reference for the importance of biotic and abiotic factors in driving the evolutionary patterns of marine plankton (line 434 in revised manuscript).

Comment: Lines 401-403. Arriving at this stage of the paper, I wonder why the author did not give any light on the rare biosphere from a microbial perspective (<https://www.nature.com/articles/ismej2016174>). I feel that the paper is heavily dominated by ideas derived from macro organisms and the author do not consider enough the existing literature is protistology.

Response: We like the suggestion to include a modern microbial perspective, and have added this point and cited the referenced paper in our manuscript (lines 343-346; line 441).

Comment: Lines 407-415. I think the authors have missed one important reference with the work of Lima-Mendez (2015; <https://www.science.org/doi/full/10.1126/science.1262073>). Trophic interactions is the main factor structuring ecosystems far ahead from abiotic factors. I think this is worth mentioning this fact at the end of the article that the rise and fall of species through time has taken place into that context. Biotic factors have been mentioned in the discussion but I think that they have not been put enough forward (But I am certainly biased).

Response: We have added the recommended reference (line 358) and further highlighted the likely importance of biotic interactions in ecosystem structure and evolutionary patterns.

Methods

Comment: I have no major criticisms on the methods since the dataset has been reused, except that this section as the rest of the paper is too long and wordy. I suggest reducing it to stick to the essential.

Response: We streamlined the Methods by removing the "Rationale" section, and removing redundancies with other sections of the text.

Reviewer #2

Opening Remarks

Trubovitz et al use the radiolarian fossil record to test for the relationship between the average (and maximum) relative abundance of a species (throughout its existence) and its longevity across the Neogene and Quaternary periods. Probing this relationship is a timely, very relevant scientific question, as many conservation efforts are decided based on species' abundance estimates. The authors make the important connection between palaeoecology and conservation biology clear in the introduction. The manuscript is well written, and the fossil dataset used is taxonomically consistent to the species level (rare in palaeoecology), with many specimens sampled per sample and very good coverage. The authors use linear regression to show that the expected positive relationship between abundance and species' persistence is not present in the radiolarian fossil record of two regions (Southern Ocean and Eastern Equatorial Pacific). The absence of a relationship between extinction risk and population abundance is intriguing, although I have some remarks about the statistical analyses and the spatial coverage of the study (see below).

Major Comments

Comment: [1] The authors' expectation of a positive relationship between abundance and longevity is based on the neutral theory of biodiversity (well introduced in the manuscript). The neutral theory is based on a stochastic demographic model (ie, all individuals regardless of species identity have same chances of dying and colonizing/giving birth). When most species are rare (which seems to be the case as relative abundances of radiolarian were "highly skewed towards low values" - lines 481-484), I would expect the relationship between abundance and longevity to be weak because there is variance around the stochastic process of death ("random walk into extinction", line 69). Thus, the difference in longevity between two species with small number individuals probably falls within this variance and is not expected to show a statistically significant difference. One way to estimate this variance would be to have a null, neutral demographic model that would constrain the expected relationship between species longevity and abundance. I understand that the computational simulation of a null model is beyond the scope of this empirically-oriented manuscript, but the variance of the stochastic process might lead to a non-linear pattern between abundance and longevity of no relationship in the low-abundance spectrum and positive relationship once species become more abundant. Moreover, given that former studies have found non-linear relationship between abundance and extinction rates (Simpson & Harnik 2009 Paleobiology [Ref. 13]: "rare and abundant genera exhibiting rates elevated over those of genera of moderate abundance"), I would expect this U-shaped pattern to also be tested with the radiolarian dataset and the final model be selected using a model selection approach (instead of visually from the plots – line 230).

Response: To address the concern that the abundance-longevity relationship might be weaker at low abundances, we performed random walk simulations with an order of magnitude difference in initial population size. The figure below shows the frequency distribution of 1000 random walk trials for hypothetical taxa with initial population sizes of 5 and 50 (arbitrary values with an order of magnitude difference). Each trial represents the mean longevity of 100 random

walk simulations at a given starting abundance, simulating the number of steps taken to reach a population size of 0 (step size= 0.1). The blue bars on the histogram below represent a stochastically derived longevity distribution for “rare” taxa (starting abundance =5), and the red bars represent a stochastically derived longevity distribution for “common” taxa (starting abundance = 50). Longevity distributions of these abundance groups are distinct, showing two clear peaks on the histogram below. This demonstrates that in a neutral scenario, we can expect a positive relationship between abundance and longevity, even at low abundances.

In response to the reviewer’s point on non-linear models, we have now added mention of residuals distributions to the manuscript and included residuals plots (Supplementary Figure 1). These confirm that a nonlinear model is unlikely to provide a better fit than a linear model. We also compared the AIC values of simple linear versus linear mixed-effects models that include biogeographic information, and found that the simple linear models had a marginally better fit to our data (further discussion of this was added to the manuscript text).

Comment: [2] Linear regressions have the underlying assumption of independent observations. The fossil dataset used, however, potentially break this assumption because, for example, the taxa are phylogenetically related, and/or the data points come from a time series that might have some type of temporal autocorrelation (eg, changes in sedimentation rate along the core). Ideally, these dependencies would be taken into account (eg, linear mixed effect model) or the authors would show that they do not affect the results (eg, there is no phylogenetic signal). Also, providing the residual plots of statistical models in the Supp Information is recommended to show that the data does not break model assumptions.

Response: To address this point, we have shown that there is no discernable phylogenetic signal (Figure 3 in revised manuscript). As requested, residuals plots for our regression analyses are now provided in Supplementary Figure 1.

Comment: [3] Former work by Harnik (Ref 5) show that geographical range, not abundance, is the main predictor of species extinction risk. Therefore, I expected the authors to discuss the geographical range of species more thoroughly in their paper. In lines 142-143 they state: “[our study] includes reasonably complete sampling of species ranges given what we know of their biogeographic distribution patterns.” However, my understanding from the methods is that the SO region was sampled in 7 locations (IODP sites) and EEP in one single site, so it is not clear to me how this spatial coverage is enough to comprise the geographical ranges of the hundreds

of studied species. Without knowing how endemic or global are these species, it is difficult to know to what degree does extinction in the IODP sites reflect global extinction of a given species. Or, even, if the IODP sites are on the edges of a species' range, its relative abundance might not be representative of its overall relative abundance (ie, across its range).

Response: We agree with this point, and have performed a new analysis investigating the impact of geographic range on longevity. While we do not know the full geographic ranges of these species, we classified each species as either "endemic" or "cosmopolitan" based on whether it was observed in both the EEP and SO, or only one of these regions. Results were assessed using boxplots and linear mixed-effects models. This analysis showed no relationship between categorical geographic range and longevity (Supplementary Figure 3). Furthermore, the mixed effects linear models produce higher AIC values than the simple linear models, indicating that they are actually worse fits of the data.

Comment: [4] Since the authors focus solely on abundance as an explanatory variable for species persistence, I expected them to have a more in-depth analysis of this variable. For example, the authors use average and maximum abundance of a species throughout its existence; however, how does the abundance of each species vary throughout its existence? How representative is the max and mean abundance of a species' across its existence? Do species decrease in abundance before they go extinct? How does the species abundance distribution (SAD) of the assemblage in each region looks like? The shape/skewness of the SAD informs us about the relative proportion of rare vs abundant species, giving insights of the expectation of the abundance-longevity relationship for each region. Does the SAD change through time? I.e., was there a moment when many rare species or many abundant species went extinct? Also, the true maximum can be sensitive to outliers, did you consider using the 90th or 95th percentile of the data instead? "Average" can sometimes be interpreted as mean, median, mode – I suggest being precise about which summary statistic you calculated.

Response: In addition to abundance as an explanatory variable for longevity, we also investigated order-level taxonomic identity and biogeographic range category as potential predictors of longevity. To address the question of abundance variability throughout species ranges, we have performed a regression analysis of standard deviation in abundance versus longevity. We found no significant relationship between these variables, indicating that variation in abundance does not predict longevity (Supplementary Figure 2).

While the reviewer raises an interesting point about species abundance distributions (SADs), we do not consider this to be within the scope of our present study. We believe that the abundance-longevity relationship should theoretically stand, regardless of SADs in each region over time. While many previous tests of neutral theory in ecological literature have focused on SADs, our goal is to focus instead on the simple prediction that common species persist longer than rare ones. For this purpose, we feel that including SADs are not necessary and would detract from the main focus of our study.

In response to the reviewer's request, we have specified that average means "mean" in the Introduction and throughout the manuscript.

As suggested, we performed a re-analysis using only the 90th percentile of relative abundance for Southern Ocean species (this dataset is the only one to include outliers with very high relative abundances). Shown below, this regression produced an R-squared of 0.0068 and a p-

value of 0.258, which agrees with our initial result that there is no relationship between abundance values and longevity (compare the scatterplot below with Figure 2b in the revised manuscript). Thus, our initial results were not strongly influenced by outliers, and we have chosen not to remove any datapoints in order to best reflect the true range of species relative abundances observed.

Comment: Table 1 and Figures 1 and 2 overlap in information. You could choose to remove the table (adding the significance information visually to the figure with dashed vs. solid lines) and show new figures that relate to a more in-depth analyses of species abundances, as suggested above. Also, Figure 3bcd and 4bcd seem to repeat information in Figure 5 (vice-versa). You could choose to show instead of boxplot (Fig 5), a violin plot where one can see more clearly the variation in the data including more info that is in the barplots (Fig 3bcd and 4bcd).

Response: We appreciate these suggestions and agree. We have removed Table 1, and added the significance information to Figure 2. We have also combined Figures 1 and 2, and Figures 3, 4, and 5 from the initial manuscript into Figure 2 and Figure 3 in the revised manuscript, respectively. Instead of a boxplot, we have re-designed Figure 3 as a violin plot with an embedded boxplot and datapoints superimposed. We have also added a new analysis of species abundances based on geographic range categories and illustrate these results in Supplementary Figure 3.

Detailed Comments

Introduction

Comment: Introduction is well written, and this comment is merely a style preference: lines 50-57 are in my opinion a bit early and too detailed for the first paragraph of the intro. I would keep the research question more general here and then come back to the details of data and specific statistical tests at the end of the intro (in the paragraph that starts at line 127).

Response: As suggested, the first paragraph now states the research question more generally, and these details have been moved to the last paragraph of the Introduction.

Comment: Line 58: this paragraph starts a bit disconnected from the one before.

Response: A transition phrase has been added (lines 55-57).

Comment: Line 59-60: add reference to this sentence.

Response: Reference added (line 59).

Comment: Line 127: I would remove “Given the inconclusive results of previous work,” - it is not completely true and, in my opinion, not necessary as the reasoning and relevance of your work is very clear already.

Response: Phrase removed.

Comment: Lines 157-160: It is good practise to provide the residuals plots of the models to show that statistical assumptions such as independence of observations are met.

Response: Residuals plots have now been added; see Supplementary Figure 1.

Methods

Comment: Line 541: affects.

Response: Changed “effects” to “affects” (line 576).

Comment: Paragraph 552: I think the discussion of relative vs. absolute abundance (or parts of it) should be in the discussion section not in the methods because it is a very important difference between contemporary and palaeo ecological data. The expectation of high abundance related to low extinction rates in ecological theory is based on absolute (not relative) abundance. So, one of the reasons you did not find a relationship could be because median/maximum relative abundance of a species is not a good proxy for its median/maximum absolute abundance.

Response: As requested, this paragraph was moved to the Discussion section (starting at line 225).

Comment: Lines 558-560: Is there a reference to this statement? And I am not sure I understood the rationale of these statements here... even if sedimentation rates are very well constrained/known, shouldn't they be considered then when calculating abundance in the sediment to try to get to a better estimate of absolute abundance? Wouldn't the way to control for an inversion of absolute vs. relative abundance be to keep sampled mass constant (instead of number of individuals)?

Response: Reference to the Neptune Database was added (line 234). Our data collection strategy was based on the number of individuals counted and the evenness of taxa in the assemblage (see Trubovitz et al. 2020); we did not track sampled mass during data collection. Our goal was to test whether relatively abundant taxa are less likely to go extinct than relatively common taxa. For this purpose, we believe that relative abundance data is the most appropriate.

Comment: Lines 565-569: Still in the same rationale paragraph, it is not clear to me why do you think the inversion would be ca. by a factor of 2 or less.

Response: We estimated that any inversion would be a factor of 2 or less, because sedimentation rates were nearly constant and varied by less than a factor of 2.

Discussion

Comment: Line 217-219: I would say “the theory *predicts* a *negative* relationship between abundance and extinction risk”. The theory itself is a metacommunity demographic model, where each individual has the same chance of dying regardless from which species it is from. So, it predicts that species with lower abundances have a higher chance of going extinct because it has less individuals to persist during the stochastic demographics of the model.

Response: Change made (line 198).

Comment: Line 244: Harnik PNAS 2011 actually shows a positive relationship between abundance and species duration. This relationship is weakened when he takes the interaction between abundance and geographical range into account. Because these two variables correlate, and range is a stronger predictor, then abundance becomes a weak predictor. Because you did not consider geographical range, I would not say your results agree with Harnik’s result.

Response: Wording was changed to reflect this point (line 283).

Comment: Line 245-246: “evolutionary behaviour” is a strange term. Prefer “evolutionary processes.”

Response: Change made (line 284).

Comment: Line 248: if extinction is not neutral, can you say something about common traits of species that went extinct or a deterministic process that selectively drove these species to extinction?

Response: Unfortunately, no. We don't know enough about species ecology to say this.

Comment: Line 273 onwards: When discussing the contrasting results between your work with those focused on marine macroinvertebrates, it would be interesting to hear the authors’ opinion about life history differences between these groups (instead of only data structure differences). Could it be that the greater dispersal capability of plankton (radiolarian) makes them more resilient to extinction than bivalve/brachiopod even when a species has low abundances? In this

sense, discussing geographic range and shifts in geographic ranges of the studied species is relevant.

Response: This topic has been added to the Discussion section (lines 337-345). The manuscript also now includes an analysis and discussion of radiolarian species geographic ranges (lines 176-195 and 418-435).

Comment: Line 291: re-phrase this sentence as you did not test how your results compare to a null model generated under the neutral assumption.

Response: "Null model" was changed to "relationship" (line 329).

Comment: Line 324: remove 'important'

Response: Changed to "many other plankton groups" (line 371).

Comment: Line 347-350: could you use your data to test this hypothesis? Do you know which species are deep vs surface-dwellers?

Response: With a few exceptions, we do not know which species were surface versus deep dwellers, so we are not able to test this hypothesis with our data.

Comment: Line 352: remove comma between subject and verb.

Response: This sentence was removed during re-organization of the Discussion section.

Comment: Line 366: "they [ecological interactions] do indeed exist" sounds strange plus this was not explicitly tested to say they "contribute significantly". I would re-write: "but the present study supports the idea that they [ecological interactions] may affect extinction dynamics over time."

Response: Sentence rewritten to: "Although the specific nature of ecological interactions among the plankton are poorly understood (particularly for extinct taxa), our results are consistent with the idea that they may contribute to extinction patterns over time" (lines 359-364).

Comment: Line 371: Remove "(at least at the level of higher taxa)" – it is confusing here, and it is made clear in line 373

Response: Change made (line 395).

Comment: Line 393: 'among' instead of 'between'? It seems there is also a lot of variation in biology and ecology *within* higher taxa, is this true? If so, this variation could explain why there was no clear difference among these higher taxa.

Response: "Between" is meant here because we are pointing out that the colonial Collodaria are different from the solitary Nassellaria and Spumellaria.

Comment: Line 410-412: This sentence should be more speculative since you did not test for life history traits, feeding strategies, interactions, tolerance. Something like: “Our results suggest that factors other than abundance, such as [...], *could have been* responsible for the extinction patterns observed in Neogene plankton.”

Response: Sentence rewritten as suggested (lines 445-448).

Comment: Re-check all reference list, as there seem to be duplicates (eg, 5 & 24, 13 & 22).

Response: Thank you, duplicates have been removed from the reference list.

Comment: Regarding the supplementary table: what do the colours mean? Some cells are coloured yellow or rose.

Response: The colors indicate origination (yellow) and extinction (red) dates and error estimates. We have added text to the Methods section explaining this (lines 492-494).

Reviewers' comments:

Reviewer #1 (Remarks to the Author):

Dear Editor, Dear Authors,

I must apologize for the delay taken to do my review.

I carefully read the response of the author and the new version of the manuscript. I feel the author responded to my comments and when they chose to not follow my recommendations they also provided solid grounds.

I still feel the manuscript is too long, especially the discussion but this is a deliberate choice of the author and I respect it.

With my best regards,

Raphael Morard

Reviewer #2 (Remarks to the Author):

Overall, I was happy with the changes done to the figures (the new Fig. 1 is beautiful), the added model residuals to the Supp Info, the text edits, and the reply to my comments (including the random walk simulation). However, I am still concerned with the conclusions of the analysis given that it is not clear how representative the relative abundance of each species averaged in the regions of SO or EPP are of the species' total average relative abundance. The hypothesis that the manuscript is testing - that species with low abundances have shorter longevities (ie, higher extinction risk) - is based on the species total abundance (total population size) and "global" longevity (i.e., species extinction, not population extirpation), but the data seems to be local(EPP)/regional(SO) estimates of species abundance and longevity.

In conservation biology, the IUCN Red List definition of extinction risk requires the description of the "scope of the assessment", which indicates whether the assessment is for the global population (total abundance of species), or if it is a regional assessment (i.e., an assessment of only part of the global population) [<https://www.iucnredlist.org/assessment/supporting-information#Population>]. The authors cite IUCN in the Introduction (Lines 41-43) and Methods (Lines 437-438) but do not make explicit in their manuscript what is their "scope of assessment" for the species they are working with. Besides potential biases in estimates of average relative abundance based on local/regional scope, this scope can also bias estimates of species' longevity.

I was surprised to see in the new biogeographical analysis how many species were categorized as "cosmopolitan", meaning that their range includes both regions (SO and EPP). For the species that have such large biogeographical ranges, the analysis of this manuscript is comparing patterns of local abundance and local longevity - thus local populational patterns. However, the hypothesis tested is based on the species-level, and the text in the manuscript comes across as species-level patterns (e.g., extinction risk, species' average/maximum abundance) although the data is local/regional. It is not clear in the manuscript, for example, how the EPP and SO abundances and longevities of a cosmopolitan species are correlated (since it occurs in both regions). Is there information on how synchronous the extinction of a radiolarian species is (i.e., do all populations of a species decrease and extirpate locally at roughly the same time, or can you have local populational extirpations in some regions, but the species remains for long periods in other regions? Also, related to abundance, how variable is the relative abundance of a species across its range?

Even species that are categorized as "endemic", might still have broad biogeographical ranges. For example, an EPP species might not be found in the SO samples (thus classified as "EPP endemic" in this study) but could still be distributed across the tropical oceans, as most low-latitude radiolarian species are pan-tropical (line 432). In this example, an "EPP endemic" species longevity and average relative abundance is then based on a single location (U1337) - so it is a very local estimate of abundance and longevity. Analysing local patterns of abundance and population persistence is interesting, but it can only be indicative of a species-level pattern of abundance and extinction if these local patterns are representative of the species' global (= total species' range) patterns. For example, the U1337 site could be at the very edge of the range of a species, where

source-sink dynamics can take place and therefore locally maintain populations in low abundances for long periods of time. Without having sampled the "source" you would find a pattern of long-lasting low-abundance populations, but because you are only sampling the "sink". As I wrote in my first review: "without knowing how endemic or global are these species, it is difficult to know to what degree extinction in the IODP sites reflects global extinction of a given species. Or, even, if the IODP sites are on the edges of a species' range, its local relative abundance might not be representative of its total, average relative abundance (i.e., across its range)." I wish I could be more positive about the new biogeographical analysis, but I am sorry to say that I am not convinced that this categorical classification allows the exploration of the relationship between longevity and biogeographical range (lines 181-182), nor that a better biogeographical radiolarian dataset is unlikely to give different results (lines 428-432).

Some ways forward, in my opinion, could be to (1) find information on each (or some) species' total average relative abundance and global longevity and relate it to the local/regional abundance and longevity in the EPP and/or SO regions; if the relationship is strong and positive, then maintaining the "species-level" scope of the manuscript can be supported, or (2) re-formulate the hypothesis, expectations and conclusions of the manuscript making explicit what is the scope of the study – that the abundance and longevity data are likely local/regional and inform us about local/regional processes of populational extirpation (i.e., local extinction, not species extinction) and local/regional relative abundance patterns (not species-level average relative abundance).

Some minor comments/thoughts:

Abstract

Line 21: add 'relative' to abundance. Since it is the first time you are stating what you did, it is important to make this clear, as modern conservation efforts are often based on absolute, not relative, abundance.

Line 26: sorry just picking this up in this second round of review, but a null model does not need to be 'realistic', it is supposed to model what would happen in the absence of the effect one is trying to test/investigate. In this sense, the neutral theory can still be a good null model, where you would expect to see a positive relationship between abundance and longevity and, by not observing it, you can then say that probably niche theory affects the observed dynamics, thus making the neutral theory still a useful, rejected, null model. I would suggest re-phrasing this sentence, to something like "These results suggest that the neutral theory does not explain/support the eco-evolutionary patterns we observe."

Line 27: "Extrinsic factors" such as? Environment? In the discussion, you discuss not only abiotic environmental change but also biotic interactions as potential factors affecting radiolarian extinction. In my understanding, "Extrinsic factors" do not account for these two factors (abiotic and biotic), as I would not say ecological interactions are "extrinsic factors".

Line 27: substitute 'behaviour' for 'dynamics' - behaviour is usually more used in biology for individual behaviour, not for population/species dynamics.

Introduction

Line 54: I would make it explicit for the reader here what you mean by biogeographic range category: "biogeographic range category (i.e., endemic or cosmopolitan)".

Line 68-70: An important comment about your hypothesis here: if you would find a relationship between longevity and (relative) abundance, niche processes could still be dominating, e.g., species with specific traits/roles have higher fitness, which leads to higher abundance, which leads to longer longevity. This idea is implicit with the "necessarily" you added in line 71, but I would

say it is too implicit. Abundance also correlates with specific traits and thus is related to longevity via niche selection.

Results

Lines 110-112: why do you mix average (mean) and average (median)? Wouldn't it be more consistent to stick to one of the two? Plus, you could just keep 'mean' and/or 'median' and remove 'average'.

Discussion

The Discussion section is very long and sometimes repetitive or very descriptive/detailed. I would try to make it shorter.

Line 222: just to clarify for a non-paleobiologist: the % here means what exactly? That 12.5% of the taxa went extinct in the last 5my; 3.2% of all taxa went extinct in the interval of 17 my (22-5 Ma)?

Lines 228-230: relative abundance may not reflect absolute abundance not only because of variation in sedimentation rate, but also because of the "zero-sum" problem among species: if species X increases greatly in absolute abundance from one sample to the other, the other species will decrease in relative abundance, even if their absolute abundances remained constant (or even increased, but to a lesser extent species X).

Lines 255-256: I do not really agree with the dichotomy here that traits are related to niche and abundance to neutral theory. If traits influence abundance, then this dichotomy is not true.

Lines 344-345: rarity is a fundamental characteristic of any species assemblage (not only microbial). The skewed species abundance distribution (ie, many rare, few abundant) is a very well know and universal pattern in community ecology.

Line 396-397: Why is the biogeographical range ~ longevity hypothesis support for the niche theory? It can be seen as a simple chance effect as well...

Methods:

Line 598: 'negative results' or non-significant results? Negative results can be confused with negative slopes.

Lines 608-612: the Methods would be clearer if this part about the Linear Mixed-Effects Models comes after the explanation of the linear regression (Line 599). I would also start a new paragraph when talking about ANOVA in Line 599.

Point-by-point responses to Reviewer #2

Major comments

Comment:

Overall, I was happy with the changes done to the figures (the new Fig. 1 is beautiful), the added model residuals to the Supp Info, the text edits, and the reply to my comments (including the random walk simulation). However, I am still concerned with the conclusions of the analysis given that it is not clear how representative the relative abundance of each species averaged in the regions of SO or EPP are of the species' total average relative abundance. The hypothesis that the manuscript is testing - that species with low abundances have shorter longevities (ie, higher extinction risk) – is based on the species total abundance (total population size) and “global” longevity (i.e., species extinction, not population extirpation), but the data seems to be local(EPP)/regional(SO) estimates of species abundance and longevity.

Response:

We acknowledge that species relative abundances do vary sometimes substantially in time and space, but we believe that our data and choice to utilize mean relative abundance as a metric is still broadly representative of overall patterns and allows us to draw meaningful conclusions. Previous literature has found that radiolarian species presence and abundance is consistent across large geographic areas. A study by Boltovskoy et al. (2010), entitled “World Atlas of Distribution of Recent Polycystina (Radiolaria)” is based on observations of biogeographic distributions of 307 extant radiolarian taxa across the Atlantic (982 samples), Indian (698 samples) and Pacific (2953 samples) ocean basins. These authors found, “None of the species covered was scarce in the Pacific but abundant in either the Atlantic or the Indian oceans.” Only 5 species were not present in all three ocean basins, and all of these were absent from the Indian ocean only (Boltovskoy et al. 2010: Table 2 and Table 3). Based on this previous work, we believe it is reasonable to assume that a single site in the tropical Pacific is generally representative of the global tropical radiolarian biome. We have also previously validated this assumption by comparing the trend of our species richness curve to the Neptune Sandbox Berlin (NSB) database, which includes 1541 samples from 26 sites in the tropical Pacific obtained from ocean drilling expeditions (Figure 2 in Trubovitz et al., 2020). Because both curves showed stability in radiolarian diversity over the last 10 Ma, this suggests that Site U1337 is not aberrant and is likely a good representative of the larger tropical radiolarian biome.

To further address this concern that species relative abundance in a local sampling area might not reflect their relative abundance throughout the entire ocean basin, we have performed a comparison of species relative abundances in two different sectors of the Southern Ocean (SO). The plot below shows mean relative abundance of species in

the Indian section of the SO as a function of mean relative abundance of the same species in the Atlantic sector of the SO. The fact that most species plot near the 1:1 line indicates a strong and significant correlation in mean relative abundance across sectors of the SO. This evidence supports the idea that radiolarian species abundances tend to be stable on the geographic scale of ocean basins.

On the potential issue of our dataset capturing global extinction versus local/regional extirpation events, we cite the well-established discipline of radiolarian biostratigraphy (thousands of studies in the modern literature). Radiolarian species presence, absence, and abundance data are among the primary microfossil methods used to assign ages to marine sediment cores and rocks (e.g., Sanfilippo et al., 1985; Gradstein et al., 2012). This discipline relies on the fact that origination and extinction events are largely synchronous in time and space; in other words, biostratigraphy would not be an effective method for dating if species appeared and disappeared in different places at substantially different times. While small variations (circa several hundred thousand years) in first and last occurrence datums are common, and to be expected on large geographic scales, these mostly do not exceed one million years and therefore should have insignificant impact on species stratigraphic longevity estimates in our study. We

expect that geographic heterogeneity in first and last occurrence datums would be below the temporal resolution of our sampling scheme in most cases. Therefore, we consider it to be a reasonable assumption that the first and last occurrences of species in our datasets mostly reflect true origination and extinction events rather than local or regional population dynamics.

To address the concern that mean relative abundance may not be a representative metric for relative abundance over time, we have plotted the relative abundance of each species observed in the SO and the eastern equatorial Pacific (EEP) over time (Appendix 1 and 2, respectively, included in the supplementary files). This dataset includes all species-level taxa observed in the EEP at IODP Site U1337, and all SO species-level taxa from sediment cores with age model quality rated “Good” or “Very Good” in the NSB database, not limited to the species with their complete stratigraphic ranges represented in the study intervals. This full dataset used to make these plots is available in the Zenodo repository: <https://doi.org/10.5281/zenodo.4014322>. These charts show that it is unusual for a species’ relative abundance to vary by more than a few percentage points over time, and the vast majority of species vary by less than 1 order of magnitude in relative abundance throughout the study interval. Thus, species that are common or rare in one sample/time interval, are similarly common or rare in most other samples/time intervals. Relative abundance is not supposed to remain constant; we would expect it to vary considerably in random walk scenarios as well as in natural assemblages at all spatial/temporal scales. Therefore, based on these expectations and the degree of variability observed in our data, we find that mean relative abundance is indeed representative and the most appropriate choice of metric for our analyses. Furthermore, the conclusions of our study are drawn from 4 orders of magnitude difference in species mean relative abundance (0.001%–13%) and 3 orders of magnitude difference in species maximum relative abundance (0.01%–45%). Thus, fluctuation in species relative abundances would have to be significantly greater than we observed for it to significantly alter our results and conclusions.

Comment:

In conservation biology, the IUCN Red List definition of extinction risk requires the description of the “scope of the assessment”, which indicates whether the assessment is for the global population (total abundance of species), or if it is a regional assessment (i.e., an assessment of only part of the global population)

[<https://www.iucnredlist.org/assessment/supporting-information#Population>]. The authors cite IUCN in the Introduction (Lines 41-43) and Methods (Lines 437-438) but do not make explicit in their manuscript what is their “scope of assessment” for the species they are working with. Besides potential biases in estimates of average relative abundance based on local/regional scope, this scope can also bias estimates of species’ longevity.

Response:

It was not our goal to assess the extinction risk of particular radiolarian species based on the IUCN guidelines. We only mention the IUCN's criteria for placing species on the Red List because our data suggest that abundance metrics may inaccurately represent extinction risk (at least for marine plankton), and we hope to bring this point to the attention of conservation biologists. We wish to convey that abundance metrics can be an oversimplification of extinction risk, due to the evidence that common species in our dataset have a history of going extinct at similar rates to rare species. Furthermore, we feel that we do make explicit the "scope of the assessment." Figure 1 clearly exhibits both the spatial and temporal scope of our study, and we frequently refer to the specific biomes and time intervals we investigated throughout the manuscript. As explained earlier in this response letter, we consider the spatial-temporal scope of our investigation to have little impact on our estimates of species longevities because these tend to be synchronous, at least at the temporal resolution relevant to this study. We agree that this issue could be a significant concern if we were attempting to estimate first/last occurrence events on the scale of hundreds–thousands of years, but in this study we are concerned with temporal precision only on the order of hundreds of thousands–millions of years.

Comment:

I was surprised to see in the new biogeographical analysis how many species were categorized as "cosmopolitan", meaning that their range includes both regions (SO and EPP). For the species that have such large biogeographical ranges, the analysis of this manuscript is comparing patterns of local abundance and local longevity - thus local populational patterns. However, the hypothesis tested is based on the species-level, and the text in the manuscript comes across as species-level patterns (e.g., extinction risk, species' average/maximum abundance) although the data is local/regional. It is not clear in the manuscript, for example, how the EPP and SO abundances and longevities of a cosmopolitan species are correlated (since it occurs in both regions). Is there information on how synchronous the extinction of a radiolarian species is (i.e., do all populations of a species decrease and extirpate locally at roughly the same time, or can you have local populational extirpations in some regions, but the species remains for long periods in other regions? Also, related to abundance, how variable is the relative abundance of a species across its range?

Response:

Although we did not specifically investigate whether abundances and longevities were correlated in the EEP and SO, we did investigate the issue of local extirpation versus global extinction in this dataset in a previous paper (Trubovitz et al., 2020). For that study, we cross-compared species lists in the SO and EEP to determine whether the species exhibiting last occurrences in the SO persisted longer at low latitudes. That analysis indicated that only 29% of species with last occurrence dates in the SO persisted at low latitudes for any amount of time, whereas 71% of them went globally extinct at the same time they disappeared from the SO. Therefore, while extirpation

might explain a small fraction of the last occurrence dates we estimated in the present study, the majority of these are likely true extinction events. Regarding the variability of relative abundance across species ranges, we have already addressed this point earlier in this response letter. Evidence supporting mean relative abundance as an abundance metric comes from the positive correlation of species mean relative abundances in two sectors of the Southern Ocean, as well as the degree of variability in relative abundance over time, as illustrated by the stratigraphic plots in Appendix 1 and 2.

Comment:

Even species that are categorized as “endemic”, might still have broad biogeographical ranges. For example, an EPP species might not be found in the SO samples (thus classified as “EPP endemic” in this study) but could still be distributed across the tropical oceans, as most low-latitude radiolarian species are pan-tropical (line 432). In this example, an “EPP endemic” species longevity and average relative abundance is then based on a single location (U1337) – so it is a very local estimate of abundance and longevity. Analysing local patterns of abundance and population persistence is interesting, but it can only be indicative of a species-level pattern of abundance and extinction if these local patterns are representative of the species’ global (= total species’ range) patterns. For example, the U1337 site could be at the very edge of the range of a species, where source-sink dynamics can take place and therefore locally maintain populations in low abundances for long periods of time. Without having sampled the “source” you would find a pattern of long-lasting low-abundance populations, but because you are only sampling the “sink”. As I wrote in my first review: “without knowing how endemic or global are these species, it is difficult to know to what degree extinction in the IODP sites reflects global extinction of a given species. Or, even, if the IODP sites are on the edges of a species’ range, its local relative abundance might not be representative of its total, average relative abundance (i.e., across its range).” I wish I could be more positive about the new biogeographical analysis, but I am sorry to say that I am not convinced that this categorical classification allows the exploration of the relationship between longevity and biogeographical range (lines 181-182), nor that a better biogeographical radiolarian dataset is unlikely to give different results (lines 428-432).

Some ways forward, in my opinion, could be to (1) find information on each (or some) species’ total average relative abundance and global longevity and relate it to the local/regional abundance and longevity in the EPP and/or SO regions; if the relationship is strong and positive, then maintaining the “species-level” scope of the manuscript can be supported, or (2) re-formulate the hypothesis, expectations and conclusions of the manuscript making explicit what is the scope of the study – that the abundance and longevity data are likely local/regional and inform us about local/regional processes of populational extirpation (i.e., local extinction, not species extinction) and local/regional relative abundance patterns (not species-level average relative abundance).

Response:

We explicitly state in the manuscript that “endemic” is very broadly classified as a species that is only present in the EEP or SO, but not both. We acknowledge that most “EEP endemic” taxa are likely present throughout all tropical ocean basins and “SO endemic” taxa may be present throughout the high latitude oceans. In the first part of this response letter, we addressed and rebutted the concern that local observations may not represent larger patterns. We presented a plot of species relative abundances in different sectors of the SO to show that a species’ mean relative abundance in one section is strongly correlated with its mean relative abundance in the other sector. Thus, we assert that it is reasonable to assume that average relative abundance in part of an ocean basin is largely representative of the whole. We also cited evidence from Boltovskoy et al. (2010) showing that the assemblage structure observed at our EEP study site are unlikely to be significantly different from other localities throughout the tropical oceans. Additional evidence from species richness trends in the NSB database also support the idea that Site U1337 is indeed representative of tropical radiolarian communities through time.

Minor comments

Line 21: add ‘relative’ to abundance. Since it is the first time you are stating what you did, it is important to make this clear, as modern conservation efforts are often based on absolute, not relative, abundance.

“Relative” added to Line 21.

Line 26: sorry just picking this up in this second round of review, but a null model does not need to be ‘realistic’, it is supposed to model what would happen in the absence of the effect one is trying to test/investigate. In this sense, the neutral theory can still be a good null model, where you would expect to see a positive relationship between abundance and longevity and, by not observing it, you can then say that probably niche theory affects the observed dynamics, thus making the neutral theory still a useful, rejected, null model. I would suggest re-phrasing this sentence, to something like “These results suggest that the neutral theory does not explain/support the eco-evolutionary patterns we observe.”

Change made. Sentence now reads “This suggests that neutral theory fails to explain the plankton ecological-evolutionary dynamics we observe.” Lines 26-27.

Line 27: “Extrinsic factors” such as? Environment? In the discussion, you discuss not only abiotic environmental change but also biotic interactions as potential factors affecting radiolarian extinction. In my understanding, “Extrinsic factors” do not account

for these two factors (abiotic and biotic), as I would not say ecological interactions are “extrinsic factors”.

We would argue that in the case of neutral theory, ecological interactions are indeed “extrinsic factors” because they are related to species traits or behavior and could therefore affect species relative abundances and longevities in ways other than the random walk posited by neutral theory.

Line 27: substitute ‘behaviour’ for ‘dynamics’ - behaviour is usually more used in biology for individual behaviour, not for population/species dynamics.

Change made; phrase in Line 27 is now “neutral dynamics.”

Introduction

Line 54: I would make it explicit for the reader here what you mean by biogeographic range category: “biogeographic range category (i.e., endemic or cosmopolitan)”.

Added “(endemic versus cosmopolitan)” to Line 54.

Line 68-70: An important comment about your hypothesis here: if you would find a relationship between longevity and (relative) abundance, niche processes could still be dominating, e.g., species with specific traits/roles have higher fitness, which leads to higher abundance, which leads to longer longevity. This idea is implicit with the “necessarily” you added in line 71, but I would say it is too implicit. Abundance also correlates with specific traits and thus is related to longevity via niche selection.

This is an interesting point on how to evaluate data where a correlation between longevity and abundance is found, which we will keep in mind should in the future we have data where such a correlation exists. As no correlation was found in our current study, this comment is moot.

Results

Lines 110-112: why do you mix average (mean) and average (median)? Wouldn't it be more consistent to stick to one of the two? Plus, you could just keep ‘mean’ and/or ‘median’ and remove ‘average’.

“Average” is now replaced with “mean” in Line 108 and Line 110. “Median” is used instead of “mean” in Lines 112 and 113 to better describe the data distribution, which has a long tail of species with very low mean relative abundances.

Discussion

The Discussion section is very long and sometimes repetitive or very descriptive/detailed. I would try to make it shorter.

We have reduced the Discussion length by approximately 25% since the last draft.

Line 222: just to clarify for a non-paleobiologist: the % here means what exactly? That 12.5% of the taxa went extinct in the last 5my; 3.2% of all taxa went extinct in the interval of 17 my (22-5 Ma)?

These extinction rates were calculated based on the widely-used “boundary crosser” method (Foote, 2000). This phrase is now clarified for a wider audience, reading “12.5% of species in each time bin went extinct over the last 5 my, compared to a 3.2% extinction rate from 22-5 Ma.” Lines 226-227.

Lines 228-230: relative abundance may not reflect absolute abundance not only because of variation in sedimentation rate, but also because of the “zero-sum” problem among species: if species X increases greatly in absolute abundance from one sample to the other, the other species will decrease in relative abundance, even if their absolute abundances remained constant (or even increased, but to a lesser extent species X).

This problem primarily arises when there are very large changes in absolute abundance between samples. We do not believe there is such large variation in our data, as explained in the Methods section (lines 532-546).

Lines 255-256: I do not really agree with the dichotomy here that traits are related to niche and abundance to neutral theory. If traits influence abundance, then this dichotomy is not true.

Sentence now changed so that there is not a dichotomy. Lines 233-236 now read, “The results of this study can be interpreted to support niche theory or neutral theory, since they show that both taxonomic identity (i.e., unique traits) and abundance (which can be related to neutral dynamics or to unique traits) were each significantly linked to

longevity.”

Lines 344-345: rarity is a fundamental characteristic of any species assemblage (not only microbial). The skewed species abundance distribution (ie, many rare, few abundant) is a very well know and universal pattern in community ecology.

We agree with this statement, but in that particular sentence we are trying to draw a parallel between paleo and present microbial communities so choose not to broaden the statement to universal patterns in community ecology.

Line 396-397: Why is the biogeographical range ~ longevity hypothesis support for the niche theory? It can be seen as a simple chance effect as well...

Because radiolarians are passively dispersed via ocean currents and the oceans are well-mixed on the time scales we are concerned with, we assume that biogeographic range is largely a reflection of species tolerance for a given environment rather than their random dispersal patterns. This idea is supported by Boltovskoy et al. (2010) and Boltovskoy and Correa (2017).

Methods:

Line 598: ‘negative results’ or non-significant results? Negative results can be confused with negative slopes.

Changed to “non-significant results” in Line 558.

Lines 608-612: the Methods would be clearer if this part about the Linear Mixed-Effects Models comes after the explanation of the linear regression (Line 599). I would also start a new paragraph when talking about ANOVA in Line 599.

Linear regressions (Lines 549-559) are discussed before linear mixed effects models (569-574). A paragraph break was added for the discussion of ANOVA in Line 560.